# Finding Landmarks of Covariate Shift with the Max-Sliced Kernel Wasserstein Distance

## Abstract

To detect distribution shifts caused by localized changes, we propose an interpretable kernel-based max-sliced Wasserstein divergence, which is computationally efficient for two-sample testing. The max landmark kernel Wasserstein distance (MLW) seeks a single data point whose kernel embedding acts as a slice of the kernel Hilbert space, such that the two samples' resulting projections (kernel evaluations between each point and the landmark) have maximal Wasserstein distance. This landmark, or multiple landmarks chosen via a greedy algorithm, provide an interpretation of localized divergences. We investigate MLW's ability to detect and localize distribution shifts corresponding to over- or -under representation of one class. Results on the MNIST and CIFAR-10 datasets demonstrate MLW's competitive statistical power and accurate landmark selection. Using the mean embedding (ME) test statistic with multiple MLW landmarks enables state-of-the-art power on the Higgs dataset.

## 1 Introduction

Covariate shift can degrade the performance of machine learning models (Quionero-Candela et al., 2009; Shimodaira, 2000) and arises when the distribution of features (also known as covariates) in the training set differs from the distribution seen in the test set during deployment. Divergence measures quantify the dissimilarity between the distributions (Sriperumbudur et al., 2009) and are essential in detecting covariate shift. Previous work (Rabanser et al., 2019; Gretton et al., 2009; Jang et al., 2022) has shown that kernel-based maximum mean discrepancy (MMD) (Gretton et al., 2012a) provides a tractable and reliable divergence method that can be directly applied to samples. Importantly, MMD can be expressed in terms of its witness function, where points with large magnitude witness function evaluations reflect a high degree of discrepancy between unnormalized kernel density estimates, enabling post-hoc interpretation of the divergence. Alternatively, the Kullback-Leibler and other $f$-divergences (Kullback & Leibler, 1951; Ali & Silvey, 1966; Rényi, 1961), are also also interpretable, in the sense that points with large magnitude log density ratio are associated with large discrepancies. However, the estimation of the underlying densities is challenging (Vapnik, 2000), and while variational optimizations (Nguyen et al., 2010; Rhodes et al., 2020) can provide estimates with bounded error when $f$ is strongly convex (Verine et al., 2023), the $f$-divergences require one distribution to be supported on the support of the second. Another consideration for the interpretation of the witness function for MMD (or the log-density ratio in $f$-divergences) is that the functions generally have multiple maxima, meaning that points with high witness function evaluations may come from disparate discrepancies spread throughout the space. Likewise, classifier-based two sample tests (Lopez-Paz & Oquab, 2016; Kübler et al., 2022; Cheng & Cloninger, 2022) are statistically powerful, but require training neural networks and identifying multiple disparate areas where the two distributions differ.

Motivated by cases where the covariate shift arises from under- or over-representation in a localized subset of the distribution (such as a mode) we propose the **max landmark kernel Wasserstein distance (MLW)**, which guarantees a localized witness function based on the kernel embedding of a single point (landmark). This builds on the max-sliced Wasserstein distance (Deshpande et al., 2019), but as in kernel-based Wasserstein distance (Zhang et al., 2020; Oh et al., 2020) and sliced versions (Wang et al., 2022; 2025), the data points are mapped to a Hilbert space. With MLW, the optimal landmark is chosen such that the two samples' resulting projections have maximal Wasserstein distance. With a translation-invariant kernel, such as

the Gaussian kernel, the landmark witness function relates to the distance from the landmark, making it easily interpretable as to where the distributions most differ. In contrast, the kernel max-sliced Wasserstein distance (Wang et al., 2022; 2025) does not ensure a localized function for interpretation.

Importantly, for strictly positive definite kernels, we prove a representer's theorem, showing that the optimal landmark is contained within the two samples. For two samples with at most $n$ points each, the optimization underlying MLW has a closed-form solution requiring $\mathcal{O}(n^2 \log(n))$ operations. For samples of size $m$ and $n$, identifying the optimal landmark simply requires finding the maximum of $m + n$ function evaluations, each with $\mathcal{O}(n \log(n))$ operations. This can be compared to the kernel-based max-sliced Wasserstein distance (Wang et al., 2022; 2025), which is an NP-Hard (Wang et al., 2025) problem requiring approximation through iterative or a semi-definite relaxation. The computational efficiency of MLW enables two types of two-sample permutation tests (Good, 2013); both generate surrogate distributions under the null hypothesis. For the first type, the divergence is calculated on each of the $B$ permutations requiring $\mathcal{O}(n^2 \log(n) + Bn^2)$ operations, since sorting only needs to be done once. For the second type, the samples are split with a fixed proportion: the first split is used to calculate the landmark, and the second split is used to calculate the test statistic and the surrogate distribution, which only requires $\mathcal{O}(n \log(n) + Bn)$ operations, and is computationally cheaper than MMD, which requires $\mathcal{O}(Bn^2)$.

MLW resembles an earlier interpretable two-sample test: the mean embedding (ME) divergence (Chwialkowski et al., 2015; Jitkrittum et al., 2016). Jitkrittum et al. (2016) proposed an algorithm to optimize the location of multiple inducing points, which are not in the sample but act as landmarks, in order to maximize a mean embedding test statistic (Chwialkowski et al., 2015). Later, Liu et al. (2020) suggested using a single data point chosen from the samples to maximize the mean embedding test statistic (ME) (Chwialkowski et al., 2015) for interpreting divergence between image datasets (we refer to this method as ME1). The contribution of MLW with respect to kernel-based max-slicing and ME is the simplicity of the optimal solution, obtained by using the sample data points as potential landmarks; the contribution with respect to ME1 is leveraging the Wasserstein distance to improve the statistical power and relevancy of the landmark. We also propose a greedy optimization of a multiple landmark version of MLW (MMLW) with a log-determinate anti-concentration penalty that follows an MMD-based algorithm proposed by Kim et al. (2016). The number of landmarks can be chosen a priori, by using a knee heuristic (Satopaa et al., 2011), or by the asymptotic ME test power (Jitkrittum et al., 2016) as a surrogate.

Following previous work (Rabanser et al., 2019; Wang et al., 2022), we test our method's ability to detect covariate shifts caused by class imbalance. This is practically relevant in assessing whether an unlabeled sample of points, seen during deployment, has the same distribution as the training set. The classes can also serve as a proxy for detecting changes in the prevalence of 'modes' of a distribution. In the case of class imbalance, we create powerful detection methods by combining MLW with either pretrained CLIP embeddings (Radford et al., 2021) or Black Box Shift Detection (BBSD) (Lipton et al., 2018), which uses the trained classifier as the representation to detect whether the source (training) and target (test) distributions are the same. Results on synthetically imbalanced real-world datasets show that MLW is competitive with MMD and ME1. In terms of the interpretation of the selected landmark, we find that the selected landmark is more consistently from the imbalanced class/mode than ME1 or the largest values of the MMD's witness function, as used in prior work (Kim et al., 2016). Specifically, on an imbalanced version of the MNIST dataset, we benchmark MLW against methods that use an optimization-based test statistic using split samples, including kernel-based approaches (Liu et al., 2020; Cheng & Xie, 2021; Wang et al., 2022; 2025) and the projected Wasserstein distance (Wang et al., 2021). Notably, MLW is a comparatively simpler approach that outperforms these methods. Finally, on the Higgs dataset, a benchmark (Chwialkowski et al., 2015) derived from high-energy physics dataset (Baldi et al., 2014), where the discrepancy between the distributions is likely not localized, MMLW has superior performance to MLW. In this case, MMLW outperforms ME with continuous-inducing point optimization (Jitkrittum et al., 2016) and the single landmark version ME1.

## 2 Preliminaries and Prior Work

Vectors are represented as boldface $\mathbf{x}$; $\mathbf{1}_n = [1, \ldots, 1]^\top$ is a column vector of $n$ ones. Element-wise operations are denoted as follows: product as $\mathbf{x} \circ \mathbf{y}$, $p$th power as $\mathbf{x}^{\circ p}$, and absolute value as $|\mathbf{x}|$. Transpose is denoted

$\mathbf{x}^\top$. Sets are represented in calligraphic font $\mathcal{X}$, as doublestruck letters $\mathbb{R}$, or, for the first $n$ natural numbers as $[n] = \{1, \ldots, n\}$. $\mathcal{X}^n = \mathcal{X} \times \cdots \times \mathcal{X}$ denotes $n$ repeated Cartesian products. A probability simplex is a set of non-negative vectors of dimension $n$ whose entries sum to one, and is denoted $\mathcal{P}_n = \{\mathbf{p} \in [0,1]^n : \sum_{i=1}^n p_i = 1\} \subset \mathbb{R}^n$. A permutation $\pi$ on $[n]$ is a bijective function $\pi : [n] \to [n]$ that can be stored as vector $\boldsymbol{\pi}$ with entries $\pi_i = \pi(i), \forall i \in [n]$; the set of permutation vectors of length-$n$ is denoted $\mathcal{Q}_n$. $\mathbb{S}^{d-1} \subset \mathbb{R}^d$ denotes the $d$-dimensional unit hypersphere such that $\mathbf{x} \in \mathbb{S}^{d-1} \implies \|\mathbf{x}\|_2 = 1$.

We consider two random variables $X, Y \in \mathcal{X}$ in a feature domain $\mathcal{X}$, distributed according to $X \sim \mu$ and $Y \sim \nu$, with $\mu, \nu \in \mathcal{P}_\mathcal{X}$, where $\mathcal{P}_\mathcal{X}$ is the set of Borel probability measures on the metric space $(\mathcal{X}, \mathfrak{d})$ defined by the distance metric $\mathfrak{d}(x, y), \forall x, y \in \mathcal{X}$. When $\mathcal{X} \subseteq \mathbb{R}^d$, we restrict our attention to those with finite moments. Let $\mathcal{P}_p(\mathbb{R}^d)$ denote the set of probability measures with finite $p$th moments. For a random vector $\mathbf{x} \sim \mu$, $d > 1$, $\mathbb{E}[\|\mathbf{x}\|_2] < \infty$ implies $\mu \in \mathcal{P}_2(\mathbb{R}^d)$, where $\|\cdot\|_2$ denotes the Euclidean norm. Given a measurable function $T : \mathcal{X} \to \mathcal{T}$, we denote the push-forward measure $\xi = T\sharp\mu$, defined as $\xi(\mathcal{A}) = T_\sharp\mu(\mathcal{A}) = \Pr(X \in \{x \in \mathcal{X} : T(x) \in \mathcal{A}\})$, $\forall \mathcal{A} \subseteq \mathcal{T}$. When $\mathcal{X}$ is discrete or $\mu$ is a discrete distribution, then the probability measure $\mu$ for the random variable $X$ is $\mu(\{x\}) = \Pr(X = x)$ such that $\mu = \sum_{x \in \text{supp}(\mu)} \mu(x)\delta_x$, where $\delta_x(\mathcal{B}) = \begin{cases} 1, & x \in \mathcal{B} \\ 0, & x \notin \mathcal{B} \end{cases}$. A finite discrete distribution corresponding to a weighted sample $\{(\mu_i, x_i)\}_{i=1}^n \in \mathcal{P}_n \times \mathcal{X}^n$ is denoted $\hat{\mu} = \sum_{i=1}^n \mu_i \delta_{x_i}$ where the masses are denoted $\boldsymbol{\mu} \in \mathcal{P}_n$ with $\mu_i = \hat{\mu}(\{x_i\})$, $\forall i \in [n]$.

## 2.1 Wasserstein Distance and Optimal Transport

The Wasserstein-$p$ distance for a metric space $(\mathcal{X}, \mathfrak{d})$ and power $p \geq 1$—assuming finite $p$th moments of $\mathfrak{d}(X, Y)$ is defined as $W_{\mathfrak{d}^p}(\mu, \nu) = \left(\inf_{\gamma \in \mathcal{P}_{\mu,\nu}} \mathbb{E}_{(X,Y) \sim \gamma}[\mathfrak{d}(X, Y)^p]\right)^{\frac{1}{p}}$, where $\mathcal{P}_{\mu,\nu}$ is the set of joint distributions with marginals $\mu$ and $\nu$. $W_{\mathfrak{d}^p}(\mu, \nu)$ is a metric between probability distributions in $\mathcal{P}_\mathcal{X}$ (Cédric, 2003; Villani, 2008; Peyré & Cuturi, 2019). For $d \geq 1$, the Euclidean metric space $(\mathbb{R}^d, \mathfrak{d}_E)$ uses $\mathfrak{d}_E(\mathbf{x}, \mathbf{y}) = \|\mathbf{x} - \mathbf{y}\|_2$.

When both distributions are finite discrete measures $\hat{\mu} = \sum_{i=1}^m \mu_i \delta_{x_i}$ and $\hat{\nu} = \sum_{i=1}^n \nu_i \delta_{y_i}$ as is the case with empirical distributions constructed from data sets, $\{(\mu_i, x_i)\}_{i=1}^m \in \mathcal{P}_m \times \mathcal{X}^m$ and $\{(\nu_i, y_i)\}_{i=1}^n \in \mathcal{P}_n \times \mathcal{X}^n$, then the Wasserstein distance can be expressed as a linear program $W_{\mathfrak{d}^p}^p(\hat{\mu}, \hat{\nu}) = \min_{\mathbf{P} \in \mathcal{P}_{\mu,\nu}} \sum_{i=1}^m \sum_{j=1}^n P_{ij} C_{ij}$, with the cost matrix $\mathbf{C} \in \mathbb{R}^{m \times n}$, $C_{ij} = \mathfrak{d}^p(x_i, y_j)$ and the feasible set $\mathcal{P}_{\boldsymbol{\mu},\boldsymbol{\nu}} = \{\mathbf{P} \in \mathcal{P}_m \times \mathcal{P}_n : \mathbf{P}\mathbf{1}_n = \boldsymbol{\mu}$ and $\mathbf{P}^\top \mathbf{1}_m = \boldsymbol{\nu}\}$. For $n \geq m$, algorithms to solve this linear program require $\mathcal{O}(n^3)$ operations (Peyré & Cuturi, 2019). In the case of samples of 1D points, the solution to the linear program can be computed efficiently after sorting, which requires only $\mathcal{O}(n \log(n))$ operations, and avoids the quadratic cost of computing the distances between all pairs of data points. Let $\boldsymbol{\pi} \in \mathcal{Q}_m$ and $\boldsymbol{\sigma} \in \mathcal{Q}_n$ denote the permutations such that $x_{\pi_1} \leq x_{\pi_2} \leq \cdots \leq x_{\pi_m}$ and $y_{\sigma_1} \leq y_{\sigma_2} \leq \cdots \leq y_{\sigma_m}$, respectively, and let $\acute{\mathbf{x}} = [x_{\pi_i}]_{i=1}^m$ and $\acute{\mathbf{y}} = [y_{\sigma_i}]_{i=1}^n$ denote the sorted samples. In the case of equal-sized samples $m = n$ with uniform masses $\boldsymbol{\mu} = \boldsymbol{\nu} = \frac{1}{m}\mathbf{1}_m$, $W_{\mathfrak{d}_E^p}^p(\hat{\mu}, \hat{\nu}) = \frac{1}{m} \sum_{i=1}^m |x_{\pi_i} - y_{\sigma_j}|^p = \frac{1}{m}\|\acute{\mathbf{x}} - \acute{\mathbf{y}}\|_p^p$. That is, after sorting, which requires $\mathcal{O}(n \log(n))$ operations, the optimal transport plan is $\acute{\mathbf{P}}^* = \frac{1}{m}\mathbf{I}$. For general masses $\boldsymbol{\mu}$ and $\boldsymbol{\nu}$, an optimal transport plan matrix $\acute{\mathbf{P}}^* \in \mathcal{P}_{\boldsymbol{\mu},\boldsymbol{\nu}}$ can be found after sorting in $\mathcal{O}(n)$ operations by the 'Northwest corner rule' dating back to Dantzig (Charnes & Cooper, 1954; Hoffman et al., 1963).

### 2.1.1 Sliced Optimal Transport

The computational efficiency of sorting versus solving a linear program has motivated slicing techniques that map data, typically vectors in $d$-dimensional Euclidean space, to one dimension. The sliced Wasserstein distance (Rabin et al., 2012; Bonneel et al., 2015; Wu et al., 2019; Deshpande et al., 2018; Kolouri et al., 2018) integrates 1D Wasserstein distances across all vectors on the unit hyper-sphere and is estimated by averaging over a finite number of random unit vectors. The max-sliced Wasserstein (Deshpande et al., 2019) seeks the single slice that maximizes the 1D divergence. Other variants include generalized slicing (Kolouri et al., 2019), distributional sliced Wasserstein distance (Nguyen et al., 2021), and energy-based sliced Wasserstein (Nguyen & Ho, 2023).

In the context of the Euclidean distance, a slice is a unit vector $\mathbf{w} \in \mathbb{S}^{d-1}$, and defines a rank-1 projection $\mathbf{w}\mathbf{w}^\top \in \mathbb{R}^{d \times d}$ to a subspace. The distance in this subspace $\mathfrak{d}_\mathbf{w}$ is equal to the distance after slicing

$$\mathfrak{d}_\mathbf{w}(\mathbf{x}, \mathbf{y}) = \|(\mathbf{w}\mathbf{w}^\top)(\mathbf{x} - \mathbf{y})\|_2 = \|\mathbf{w}\|_2 |\mathbf{w}^\top(\mathbf{x} - \mathbf{y})| = |\mathbf{w}^\top\mathbf{x} - \mathbf{w}^\top\mathbf{y}| = \mathfrak{d}_\mathrm{E}(\mathbf{w}^\top\mathbf{x}, \mathbf{w}^\top\mathbf{y}).$$

For $\boldsymbol{\mu} = \boldsymbol{\nu} = \frac{1}{m}\mathbf{1}_m$, $W^p_{\mathfrak{d}_\mathbf{w}}(\hat{\mu}, \hat{\nu}) = \frac{1}{m}\sum_{j=1}^m |\mathbf{w}^\top\mathbf{x}_{\pi_j} - \mathbf{w}^\top\mathbf{y}_{\sigma_j}|^p$, where the permutations $\boldsymbol{\pi}$ and $\boldsymbol{\sigma}$ ensure that $\mathbf{w}^\top\mathbf{x}_{\pi_1} \leq \mathbf{w}^\top\mathbf{x}_{\pi_2} \leq \cdots \leq \mathbf{w}^\top\mathbf{x}_{\pi_m}$ and $\mathbf{w}^\top\mathbf{y}_{\sigma_1} \leq \mathbf{w}^\top\mathbf{y}_{\sigma_2} \leq \cdots \leq \mathbf{w}^\top\mathbf{y}_{\sigma_m}$, respectively. The empirical max-sliced Wasserstein-$p$ distance is $MSW_p(\hat{\mu}, \hat{\nu}) = \sup_{\mathbf{w} \in \mathbb{S}^{d-1}} W^p_{\mathfrak{d}_\mathbf{w}}(\hat{\mu}, \hat{\nu})$. Statistically, sample complexity for MSW, comparing $MSW_p(\mu, \nu)$ and $MSW_p(\hat{\mu}, \hat{\nu})$, has been studied extensively (Xi & Niles-Weed, 2022; Goldfeld et al., 2024; Boedihardjo, 2025). However, in practice obtaining the max-slice is itself a difficult optimization even for discrete measures.

### 2.1.2 Max-sliced Kernel Divergences

The kernel trick is a well-studied approach to create non-linear functions as linear parametrizations of data embedded in a reproducing kernel Hilbert space (RKHS). Given a symmetric (real-valued) positive semi-definite kernel function $\kappa : \mathcal{X} \times \mathcal{X} \to \mathbb{R}$, then $\kappa$ uniquely defines a reproducing kernel Hilbert space $\mathcal{H}$, with inner products denoted $\langle \cdot, \cdot \rangle$, and satisfies two key properties (Scholkopf & Smola, 2001): $\forall x \in \mathcal{X}, \quad \kappa(\cdot, x) \in \mathcal{H}$ and $\forall x \in \mathcal{X}, h \in \mathcal{H}, \quad h(x) = \langle h, \kappa(\cdot, x) \rangle$. The properties can also expressed in terms of embedding function $\phi : \mathcal{X} \to \mathcal{H}$: 1) $\forall x \in \mathcal{X} \; \phi(x) = \kappa(\cdot, x) \in \mathcal{H}$, and 2) $\forall h \in \mathcal{H}, h(x) = \langle h, \phi(x) \rangle$. Additionally, $\forall x, y \in \mathcal{X}, \langle \phi(x), \phi(y) \rangle = \kappa(x, y)$.

In the RKHS, a slice $\omega \in \{\omega \in \mathcal{H} \; : \; \|\omega\|_\mathcal{H} = 1\}$ defines a rank-1 projection operator $\omega \otimes \omega \in \mathcal{H} \times \mathcal{H}$ to a one-dimensional subspace in the RKHS. The sliced kernel-induced distance is $\mathfrak{d}_\omega(x, y) = \|(\omega \otimes \omega)(\phi(x) - \phi(y))\|_\mathcal{H} = \|\omega\|_\mathcal{H} |\langle \omega, \phi(x) - \phi(y) \rangle| = |\omega(x) - \omega(y)|$, where $\omega(x)$ and $\omega(y)$ are real-valued function evaluations of the slicing function. The kernel max-sliced Wasserstein distance (KMS) (Wang et al., 2025),[1] can be expressed in terms of witness functions as $KMS_{\kappa,p}(\mu, \nu) = \sup_{\omega \in \mathcal{F}} W^p_{\mathfrak{d}_\omega}(\mu, \nu) = \sup_{\omega \in \mathcal{H} \text{ s.t. } \|\omega\|_\mathcal{H} = 1} W^p_{\mathfrak{d}_\omega}(\mu, \nu)$, where $\mathcal{F} = \{h \in \mathcal{H} : \|h\|_\mathcal{H} \leq 1\}$. This equality holds because the optimal witness function will always be unit-norm due to the Wasserstein-$p$ distance being homogeneous. Intuitively, optimizing the witness function to maximize the Wasserstein distance of the pushforward distribution of the witness function evaluations, $KMS_{\kappa,p}(\mu, \nu) = \sup_{\omega \in \mathcal{F}} W^p_{\mathfrak{d}_\mathrm{E}}(\omega \sharp \mu, \omega \sharp \nu)$, should be more powerful than comparing just the means as in MMD. However, unlike MMD there is no closed-form solution for the optimal witness function as $KMS_{\kappa,p}(\mu, \nu) = \sup_{\omega \in \mathcal{F}} W^p_{\mathfrak{d}_\mathrm{E}}(\omega \sharp \mu, \omega \sharp \nu) = \sup_{\omega \in \mathcal{F}} \left( \inf_{\gamma \in \mathcal{P}_{\mu,\nu}} \mathbb{E}_{(X,Y) \sim \gamma} [|\omega(X) - \omega(Y)|^p] \right)^{\frac{1}{p}}$ is a difficult saddlepoint optimization (NP-Hard (Wang et al., 2025)), as in the original max-sliced Wasserstein distance (Deshpande et al., 2019). Additionally, like MMD the witness function is not localized.

In the context of interpreting deep kernels, Liu et al. (2020) suggested selecting a point from the two equal-sized samples that maximizes the mean embedding (ME) test statistic (Chwialkowski et al., 2015). We refer to this as the ME1 test (see Listing 1):

$$ME1_\kappa(\hat{\mu}, \hat{\nu}) = \max_{z \in \{x_i\}_{i=1}^m \cup \{y_i\}_{i=1}^n} \frac{n \bar{\delta}_z^2}{\frac{1}{n-1}\sum_{i=1}^n (\delta_i(z) - \bar{\delta}_z)^2}, \quad \bar{\delta}_z = \frac{1}{n}\sum_{i=1}^n \delta_i(z), \quad \delta_i(z) = \kappa(x_i, z) - \kappa(y_i, z). \quad (1)$$

When ME1 is used in a permutation test it requires $\mathcal{O}(n^2)$ operations for each permutation. In a split permutation test, after the selection of the landmark $z^*$, it only requires $\mathcal{O}(n)$ operations per permutation.

## 3 Proposed Methodology

To achieve an efficient and interpretable kernel max-sliced Wasserstein distance, we propose to limit the witness function to those defined by the Hilbert-space embedding of the sample data points. As the data points that have maximal divergence are useful guides in identifying localized discrepancies, we refer to them as landmarks. The algorithm to compute the distance and find the landmark requires evaluating the one-dimensional Wasserstein distance for the set of candidate landmarks formed by the union of the two

---

[1]The kernel max-sliced Wasserstein distance was studied in the context of vector-valued RKHS (Micchelli & Pontil, 2005) as the kernel projected Wasserstein distance (KPM) (Wang et al., 2022).

samples and then taking the maximum. The evaluation for each candidate landmark requires computing a one-dimensional Wasserstein distance $\mathcal{O}(n \log(n))$ with an overall computational complexity of $\mathcal{O}(n^2 \log(n))$ for samples of size $n$. We also propose an algorithm to greedily select multiple landmarks.

### 3.1 Max Landmark Kernel Wasserstein Distance

To ensure witness functions are unit-norm, we restrict our attention to the witness functions corresponding to the embeddings from normalized kernels $\kappa(x,y) = \langle \phi(x), \phi(y) \rangle$ such that $\|\phi(z)\|_{\mathcal{H}} = 1 \quad \forall z \in \mathcal{X}$. An unnormalized kernel $\kappa'$ with $\kappa'(x,x) > 0, \forall x \in \mathcal{X}$, can be be normalized as $\kappa(x,y) = \frac{\kappa'(x,y)}{\sqrt{\kappa'(x,x)\kappa'(y,y)}}$. We refer to a point $z \in \mathcal{X}$ defining the witness function $\omega_z = \phi(z) \in \mathcal{H}$ as a **landmark**. Note that $\omega_z$ is a slice in the RKHS. The landmark kernel-induced distance, $\mathfrak{d}_{\phi(z)}(x,y) = \mathfrak{d}_{\mathrm{E}}(\kappa(z,x), \kappa(z,y)) = |\kappa(z,x) - \kappa(z,y)|$ compares the points $x, y$ in terms of their kernel similarity to the landmark $z$. The set of witness functions for candidate landmarks as $\mathcal{L} = \{\phi(z) \in \mathcal{H} : z \in \mathcal{X}\} \subset \{\omega \in \mathcal{H} : \|\omega\|_{\mathcal{H}} = 1\} \subset \{\omega \in \mathcal{H} : \|\omega\|_{\mathcal{H}} \leq 1\} = \mathcal{F}$. For $z \in \mathcal{X}$ with $\omega = \phi(z)$, the landmark-sliced kernel Wasserstein (LSKW) distance is

$$LSKW_{z,\kappa,p}(\mu,\nu) = W_{\mathfrak{d}_{\phi(z)}^p}(\mu,\nu) = W_{\mathfrak{d}_{\omega_z}^p}(\mu,\nu) = W_{\mathfrak{d}_{\mathrm{E}}^p}(\omega_z \sharp \mu, \omega_z \sharp \nu), \tag{2}$$

and the max landmark kernel Wasserstein (MLW) distance is

$$MLW_{\kappa,p}(\mu,\nu) = \sup_{z \in \mathcal{X}} W_{\mathfrak{d}_{\phi(z)}^p}(\mu,\nu) = \sup_{\omega_z \in \mathcal{L}} W_{\mathfrak{d}_{\omega_z}^p}(\mu,\nu). \tag{3}$$

Clearly, MLW is a lower bound on the KMS distance $MLW_{\kappa,p}(\mu,\nu) \leq KMS_{\kappa,p}(\mu,\nu)$ and is a probability distance metric for characteristic and normalized kernels (proof in Appendix A.1).

**Theorem 1.** *If $\kappa$ is characteristic (Fukumizu et al., 2008) (or universal (Micchelli et al., 2006)) and normalized $\kappa(z,z) = 1, \ \forall z \in \mathcal{X}$, then $MLW_{\kappa,p}(\mu,\nu)$ is a probability distance metric for $\mu, \nu, \xi \in P(\mathcal{X})$ with finite pth moments.*

While Theorem 1 demonstrates that seeking landmark slices across the whole space yields a probability metric, in practice for finite discrete measures, we restrict landmarks to points in the support $\mathcal{Z} = \mathrm{supp}(\hat{\mu}) \cup \mathrm{supp}(\hat{\nu}) = \{z_i\}_{i=1}^l = \{x_i\}_{i=1}^m \cup \{y_i\}_{i=1}^n \subseteq \mathcal{X}$. In this case, $\kappa$ needs to be strictly positive definite, such that embeddings of any discrete set of distinct data points are linearly independent, to ensure that the simple discrete optimization over the set of $l = m + n$ possible landmarks is optimal.

**Theorem 2.** *If $\kappa$ is strictly positive definite and normalized $\kappa(z,z) = 1, \ \forall z \in \mathcal{X}$, then*

$$MLW_{\kappa,p}(\hat{\mu},\hat{\nu}) = \sup_{z \in \mathcal{X}} W_{\mathfrak{d}_{\phi(z)}^p}(\hat{\mu},\hat{\nu}) = \max_{z \in \mathcal{Z}} W_{\mathfrak{d}_{\phi(z)}^p}(\hat{\mu},\hat{\nu}) = \max\{W_{\mathfrak{d}_{\phi(z_1)}^p}(\hat{\mu},\hat{\nu}), \ldots, W_{\mathfrak{d}_{\phi(z_l)}^p}(\hat{\mu},\hat{\nu})\}. \tag{4}$$

The proof is in Appendix A.1. Note that the Gaussian kernel, other radial basis functions, and bounded continuous translation-invariant kernels that are strictly positive definite are also characteristic (Sriperumbudur et al., 2011). For positive semidefinite kernels, $\max_{z \in \mathcal{Z}} W_{\mathfrak{d}_{\phi(z)}^p}(\hat{\mu},\hat{\nu}) \leq \sup_{z \in \mathcal{X}} W_{\mathfrak{d}_{\phi(z)}^p}(\hat{\mu},\hat{\nu})$.

For two samples, MLW's witness function is $\omega_{z^*} = \arg\max_{\omega_z \in \mathcal{L}} \min_{\mathbf{P} \in \mathcal{P}_{\mu,\nu}} \sum_i \sum_j P_{ij}|\omega_z(x_i) - \omega_z(y_j)|^p$, which can be contrasted to MMD's witness function $\omega_{z^*} = \arg\max_{\omega_z \in \mathcal{F}} \frac{1}{m} \sum_i \omega_z(x_i) - \frac{1}{n} \sum_j \omega_z(y_j)$ in that MLW restricts the witness function to landmark embeddings, $\omega_{z^*}(\cdot) = \kappa(\cdot, z^*)$, and uses the Wasserstein distance as the objective rather than the difference of the means. In the context of finite sample guarantees for MLW, we note that the Rademacher complexity of $\mathcal{L} \subset \mathcal{F}$ for sample of size $n$ is $\mathfrak{R}_n(\mathcal{L}) \leq \mathfrak{R}_n(\mathcal{F}) \leq \frac{1}{\sqrt{n}}$, which follows from the facts that the Rademacher complexity is monotone with respect to set inclusion and the Rademacher complexity of the the unit-ball in the RKHS is $\mathfrak{R}_n(\mathcal{F}) \leq \frac{1}{\sqrt{n}}$. Likewise, this observation leads to an upper bound on the sample complexity for MLW of $\mathcal{O}(n^{-1/(2p)})$, which is dimension free, following from the results in Boedihardjo (2025) and Wang et al. (2025, Theorem 3.2); however, there remains dependence between the dimension and the sample size for two-sample testing as noted for MMD (Ramdas et al., 2015).

#### 3.1.1 Efficient Computation of MLW

As clear from equation 4, the selection of the optimal landmark is a discrete maximization and the main computational cost of MLW is the computation of the LSKW distances, which requires sorting the $l$ columns

of the kernel matrices $\mathbf{K}_{XZ} = [\kappa(x_i, z_k)]_{i=1,k=1}^{m,\,l} = [\mathbf{k}_{Xz_k}]_{k=1}^l$ and $\mathbf{K}_{YZ} = [\kappa(y_i, z_k)]_{i=1,k=1}^{n,\,l} = [\mathbf{k}_{Yz_k}]_{k=1}^n$. In the case of equal sample sizes $m = n$ and uniform masses $\mu_1 = \cdots = \mu_m = \nu_1 = \cdots = \nu_m = \frac{1}{m}$,

$$MLW_{\kappa,p}^p(\hat\mu, \hat\nu) = \max_{z \in \mathcal{Z}} W_{\mathfrak{d}_{\phi(z)}^p}^p(\hat\mu, \hat\nu) = \max_{k \in [l]} \frac{1}{m} \sum_{i=1}^m |\kappa(x_{\pi_i^k}, z_k) - \kappa(y_{\sigma_i^k}, z_k)|^p = \max_{k \in [l]} \frac{1}{m} \|\acute{\mathbf{k}}_{Xz_k} - \acute{\mathbf{k}}_{Yz_k}\|_p^p, \quad (5)$$

where for $k \in [l]$, $\boldsymbol{\pi}^k$ and $\boldsymbol{\sigma}^k$ denotes the permutations such that $\kappa(x_{\pi_1^k}, z_k) \leq \cdots \leq \kappa(x_{\pi_m^k}, z_k)$ and $\kappa(y_{\sigma_1^k}, z_k) \leq \cdots \leq \kappa(y_{\sigma_n^k}, z_k)$, $\acute{\mathbf{k}}_{Xz_k} = [\kappa(x_{\pi_1^k}, z_k), \ldots, \kappa(x_{\pi_m^k}, z_k)]^\top$ and $\acute{\mathbf{k}}_{Yz_k} = [\kappa(y_{\sigma_1^k}, z_k), \ldots, \kappa(y_{\sigma_n^k}, z_k)]^\top$ are the vectors of sorted kernel evaluations. We note that the optimal landmark is not guaranteed to be unique with the set of optimal landmarks $\{k : W_{\mathfrak{d}_{\phi(z_k)}^p}(\hat\mu, \hat\nu) = \max_{z \in \mathcal{Z}} W_{\mathfrak{d}_{\phi(z)}^p}(\hat\mu, \hat\nu)\}$. Assuming no ties, the optimal landmark $z^* = \arg\max_{z \in \mathcal{Z}} W_{\mathfrak{d}_{\phi(z)}^p}(\hat\mu, \hat\nu)$ is $z^* = z_{k^*}$ where $k^* = \arg\max_{k \in [l]} \frac{1}{m} \|\acute{\mathbf{k}}_{Xz_k} - \acute{\mathbf{k}}_{Yz_k}\|_p^p$. A minimal implementation using NumPy (Harris et al., 2020) is given in Listing 2.

For general masses or sample sizes, the MLW can be expressed in terms of the optimal transport plan $\acute{\mathbf{P}}^k$ obtained via the Northwest corner rule for the weights sorted in terms of the $k$th candidate landmark (see Appendix A.2). For uniform masses $\boldsymbol{\mu} = \frac{1}{m}\mathbf{1}_m$ and $\boldsymbol{\nu} = \frac{1}{n}\mathbf{1}_n$, but possibly non-equal sample sizes $m$ and $n$, the optimal transport solution for all landmarks $\acute{\mathbf{P}}^1 = \cdots = \acute{\mathbf{P}}^{m+n} = \acute{\mathbf{P}}$ is the same and does not depend on the data, which means the Northwest corner rule needs to be computed once. The optimality of a data-independent solution for uniform masses enables efficient computation of the surrogate distribution required for the two-sample permutation test (see Appendix A.3).

## 3.2  Two-Sample Permutation Test

Given two samples of independent data points $x_1, \ldots, x_m$ and $y_1, \ldots, y_n$ coming from unknown distributions $\mu$ and $\nu$ respectively, we test the null hypothesis $H_0 : \mu = \nu$ versus the alternative hypothesis $H_1 : \mu \neq \nu$. Using the max landmark kernel Wasserstein distance as a test statistic for comparing two samples, we perform this hypothesis test via a permutation test, generating a surrogate null distribution of the test statistic by permuting data points between the samples (Good, 2013). Let $D = MLW_{\kappa,p}^p(\hat\mu, \hat\nu)$ denote the original test statistic and $\tilde{D}^b$ denote the test statistic for the $b$th permutation. The p-value is the proportion $\{\tilde{D}^b\}_{b=1}^B$ greater than or equal to $D$, $p_{\text{value}} = \frac{|\{b : \tilde{D}^b \geq D, \quad b \in [B]\}|}{B}$. Given a user-defined significance level $\alpha$, the null hypothesis is rejected if $p_{\text{value}} \leq \alpha$. Calculating the divergence across $B$ random permutations requires only a single sort of each of columns of the pooled kernel matrix (see Appendix A.3). Across $B$ permutations, the entire procedure requires $\mathcal{O}(n^2 + n^2 \log(n) + Bn^2) = \mathcal{O}(n^2 \log(n) + Bn^2)$ operations, which is on the same order as MMD when $\log(n)$ of the same order as $B$, which is the typical case.

### 3.2.1  Split Permutation Test for MLW

An alternative approach, which follows two sample tests (Gretton et al., 2012b; Jitkrittum et al., 2016; Liu et al., 2020; Wang et al., 2022), is to split each of the two samples into a training set and a test set. Without loss of generality, assume the training set for the first sample is the first $m^{\text{Tr}}$ points, $\{x_i^{\text{Tr}}\}_{i=1}^{m^{\text{Tr}}} = \{x_i\}_{i=1}^{m^{\text{Tr}}}$ and the test set is the remainder $\{x_i^{\text{Te}}\}_{i=1}^{m^{\text{Te}}} = \{x_i\}_{i=1+m^{\text{Tr}}}^m$, and likewise, $\{y_i^{\text{Tr}}\}_{i=1}^{n^{\text{Tr}}} = \{y_i\}_{i=1}^{n^{\text{Tr}}}$ and $\{y_i^{\text{Te}}\}_{i=1}^{n^{\text{Te}}} = \{y_i\}_{i=1+n^{\text{Tr}}}^n$. The empirical distributions of the training and test sets are then $\hat\mu_{\text{Tr}} = \frac{1}{m^{\text{Tr}}} \sum_i \delta_{x_i^{\text{Tr}}}$ and $\hat\nu_{\text{Tr}} = \frac{1}{n^{\text{Tr}}} \sum_i \delta_{y_i^{\text{Tr}}}$, and $\hat\mu_{\text{Te}} = \frac{1}{m^{\text{Te}}} \sum_i \delta_{x_i^{\text{Te}}}$ and $\hat\nu_{\text{Te}} = \frac{1}{n^{\text{Te}}} \sum_i \delta_{y_i^{\text{Te}}}$, respectively. The pooled sample for training is $\{z_i^{\text{Tr}}\}_{i=1}^{l_{\text{Tr}}} = \{x_i^{\text{Tr}}\}_{i=1}^{m^{\text{Tr}}} \cup \{y_i^{\text{Tr}}\}_{i=1}^{n^{\text{Tr}}}$, with $l_{\text{Tr}} = m^{\text{Tr}} + n^{\text{Tr}}$. The optimal landmark (for simplicity, we assume ties are broken arbitrarily) is then obtained based on the two training sets, then MLW is calculated on the test sets. In this case, a permutation test involves creating a surrogate null distribution by shuffling the test sets and does not require identifying a new landmark. The landmark on the training set and the test statistic are $z^* = \arg\max_{z \in \{z_i^{\text{Tr}}\}_{i=1}^{m^{\text{Tr}}}} W_{\mathfrak{d}_{\phi(z)}^p}(\hat\mu_{\text{Tr}}, \hat\nu_{\text{Tr}}), \quad D_{\text{Te}} = W_{\mathfrak{d}_{\phi(z^*)}^p}(\hat\mu_{\text{Te}}, \hat\nu_{\text{Te}})$. Assuming the split is a fraction of the samples, the computation of the landmark requires $\mathcal{O}(n^2 \log n)$ operations, but the test statistic requires only $\mathcal{O}(n \log n)$ operations, and only the landmark needs to be kept from the training set. During the permutation test, the sorting can be done once for the pooled test sample. The computational complexity of the permutation test is $\mathcal{O}(n \log n + Bn)$, which is less than the $\mathcal{O}(Bn^2)$ required by MMD.

### 3.3 Multiple Max Landmark Wasserstein Distance (MMLW)

A limitation of MLW is that it selects a single data point as a reference landmark to describe the discrepancy; thus, when multiple discrepancies exist only one will be identified, leading to an incomplete understanding of distribution shift. To identify multiple non-redundant landmarks, we follow the approach described in Kim et al. (2016) that greedily selects multiple landmarks. Subsequent landmarks are greedily selected by choosing the points with the largest magnitude evaluations from the MMD witness function, combined with a log-determinant on the kernel between landmarks as an anti-concentration penalty. For MLW, the objective for selecting $q$ landmarks is

$$\sup_{\mathcal{Z}_q \subset \mathcal{Z}} \sum_{z \in \mathcal{Z}_q} W_{\mathfrak{d}^p_{\phi(z_q)}}(\hat{\mu}, \hat{\nu}) + \log \det K_{\mathcal{Z}_q}, \tag{6}$$

where $K_{\mathcal{Z}_q} = [\kappa(z, z')]_{z, z' \in \mathcal{Z}_q}$ is the kernel matrix for a set of landmarks. A greedy solution $\mathcal{Z}_q^* = \{z_{k_1}, \ldots, z_{k_q}\}$ is given by the sequence of indices $k_1, \ldots, k_q$, where $k_1 = k^* \in \arg\max_{k \in [l]} W_{\mathfrak{d}^p_{\phi(z_k)}}(\hat{\mu}, \hat{\nu})$, $k_2 \in \arg\max_{k \in [l]} W_{\mathfrak{d}^p_{\phi(z_k)}}(\hat{\mu}, \hat{\nu}) + \log \det K_{\{z_{k_1}, z_k\}}$, and $k_q \in \arg\max_{k \in [l]} W_{\mathfrak{d}^p_{\phi(z_k)}}(\hat{\mu}, \hat{\nu}) + \log \det K_{\{z_{k_1}, \ldots, z_{k_{q-1}}, z_k\}}$. Like in the MMD case (Kim et al., 2016), the greedy approach is guaranteed to find a solution that is within a $(1 - 1/e)$ factor of the optimal solution. This performance guarantee holds because the objective function is both monotone and submodular, properties it inherits from its constituent parts: the sum of the non-negative Wasserstein distances is monotone and modular, while the log-determinant of the kernel matrix is well-known to be a monotone and submodular function.[2] To avoid performing this greedy optimization in each permutation when generating the surrogate distribution, we only consider MMLW in the split permutation test setting. Listing 5 and Listing 6 provides minimal NumPy implementations.

The MMLW test statistic $\sum_{z \in \{z_{k_1}, \ldots, z_{k_q}\}} W_{\mathfrak{d}^p_{\phi(z)}}(\hat{\mu}, \hat{\nu})$ may also be more powerful, just as the distributional sliced Wasserstein distance (Nguyen et al., 2021) and energy-based sliced Wasserstein distance (Nguyen & Ho, 2023) are more powerful than either max-sliced or average. However, the value of $q$ must be selected appropriately as a larger $q$ than necessary may decrease power. The value of $q$ can be chosen by performing cross-validation on the training split of the two samples but this adds computational complexity. Alternatively, a heuristic "knee" method such as Kneedle (Satopaa et al., 2011) can be used to find a point of diminishing returns for the cumulative sum of the divergences. We apply the Kneedle algorithm with the 'concave' and 'increasing' settings to the sequence $W_{\mathfrak{d}^p_{\phi(z_{k_1})}}(\hat{\mu}, \hat{\nu}), \sum_{i=1}^{2} W_{\mathfrak{d}^p_{\phi(z_{k_i})}}(\hat{\mu}, \hat{\nu}), \ldots, \sum_{i=1}^{q} W_{\mathfrak{d}^p_{\phi(z_{k_i})}}(\hat{\mu}, \hat{\nu})$.

A more statistical approach is to use the asymptotic test power of the ME test statistic (Jitkrittum et al., 2016) to select the number of landmarks. Both MMLW-knee and MMLW-ME approaches are options in Listing 5. The computational complexity of the greedy algorithm is an additional $\mathcal{O}(nq^3)$ after the $\mathcal{O}(n^2 \log(n))$ of computing all the LSKW values. Thus, it is practically the same when $q \leq n^{1/3}$. The use of the ME test statistic for selecting the number of landmarks motivates directly using the ME test statistic with the landmarks selected by MMLW.

## 4 Experimental Results

We compare the ability of MLW and MMLW to MMD, ME1, and other baselines to detect shifts in distribution and to interpret these shifts. Results for MLW and MMLW generally use $p = 2$, and results for kernel-based approaches generally use a Gaussian kernel $\kappa(x, y) = e^{-\frac{\|x-y\|^2}{2\sigma^2}}$, where following the median heuristic, $\sigma$ is the median of the pairwise distances among all points in the pooled sample. Quantitative results of the two-sample hypothesis testing are in the form of statistical power tests (the probability of correctly rejecting the null hypothesis). The interpretability of MLW is assessed by examining the landmarks instances in datasets with known covariate shifts.

### 4.1 Toy Data (1D)

To illustrate MLW we consider a one-dimensional toy example where one sample is from a standard Gaussian and the second is from a mixture involving outlier distributions with one mode or two modes with different

---

[2]In comparison, the ME test statistic (Jitkrittum et al., 2016) is not submodular.

variances as shown in Figure 1, wherein the absolute value of MMD's witness function is plotted along with the LKSW-2 distance (the curves under the null distribution $\mu = \nu$ are shown in Figure 3 in the appendix).

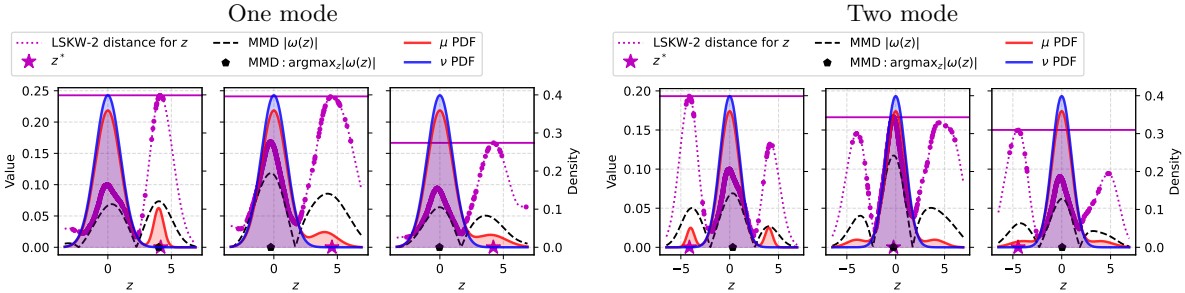

Figure 1: Illustration of MLW and MMD witness functions for toy example when $\mu = \mathcal{N}(0, 1)$ and $\nu = 0.9\mu + 0.1\xi$, where the outlier distribution has either one mode $\xi = \mathcal{N}(4, \sigma^2)$ or two modes $\xi = 0.5\mathcal{N}(-4, \sigma^2) + 0.5\mathcal{N}(4, \sigma^2)$, tested across $\sigma^2 \in \{0.15, 1, 1.15\}$ with $m = n = 250$. The purple dots and dotted curves show the landmark sliced kernel Wasserstein-2 distance (LSKW-2) as a function of $z \in \mathcal{X} = \mathbb{R}$ (star); the dashed line is the absolute value of the MMD witness function whose maximum is indicated by hexagon; and the solid blue and red lines show the true PDF.

Table 1 shows that MLW has nearly 100% power at $m = n = 250$ at a significance level of $\alpha = 0.05$, using $B = 100$ permutations, where power is calculated across 100 trials and results report average and standard deviation for 5 runs.

Table 1: Statistical power for toy examples when $\mu = \mathcal{N}(0, 1)$ and $\nu = 0.9\mu + 0.1\xi$ for $m = n = 250$, significance level of $\alpha = 0.05$, average and standard deviation across 5 runs using 100 trials each.

| | One mode $\xi = \mathcal{N}(4, \sigma^2)$ | | | Two mode $\xi = 0.5\mathcal{N}(-4, \sigma^2) + 0.5\mathcal{N}(4, \sigma^2)$ | | |
| | $\sigma^2 = 0.15$ | $\sigma^2 = 1$ | $\sigma^2 = 1.15$ | $\sigma^2 = 0.15$ | $\sigma^2 = 1$ | $\sigma^2 = 1.15$ |
|---|---|---|---|---|---|---|
| MMD | $0.894 \pm 0.020$ | $0.762 \pm 0.046$ | $0.738 \pm 0.029$ | $0.740 \pm 0.026$ | $0.598 \pm 0.044$ | $0.538 \pm 0.063$ |
| MLW | $1.000 \pm 0.000$ | $1.000 \pm 0.000$ | $1.000 \pm 0.000$ | $1.000 \pm 0.000$ | $0.990 \pm 0.006$ | $0.988 \pm 0.007$ |
| MLW-split | $0.994 \pm 0.008$ | $0.982 \pm 0.022$ | $0.976 \pm 0.019$ | $0.734 \pm 0.038$ | $0.668 \pm 0.028$ | $0.636 \pm 0.040$ |
| MMLW, $q = 2$ | $0.976 \pm 0.010$ | $0.946 \pm 0.034$ | $0.922 \pm 0.019$ | $0.938 \pm 0.019$ | $0.830 \pm 0.014$ | $0.782 \pm 0.040$ |
| MMLW, $q = 3$ | $0.934 \pm 0.020$ | $0.898 \pm 0.020$ | $0.880 \pm 0.031$ | $0.950 \pm 0.014$ | $0.866 \pm 0.014$ | $0.840 \pm 0.011$ |
| MMLW, $q = 4$ | $0.894 \pm 0.038$ | $0.868 \pm 0.030$ | $0.828 \pm 0.029$ | $0.922 \pm 0.021$ | $0.832 \pm 0.020$ | $0.790 \pm 0.021$ |
| MMLW-knee | $0.988 \pm 0.007$ | $0.952 \pm 0.021$ | $0.932 \pm 0.019$ | $0.914 \pm 0.010$ | $0.814 \pm 0.023$ | $0.764 \pm 0.027$ |
| MMLW-ME | $0.974 \pm 0.012$ | $0.942 \pm 0.035$ | $0.926 \pm 0.030$ | $0.918 \pm 0.007$ | $0.834 \pm 0.026$ | $0.790 \pm 0.029$ |

The table also has results for MLW-split, MMLW with different number of landmarks, and MMLW with the number of landmarks selected by either the knee method or the power estimate for the ME test statistic. MLW-split has higher power with MMD, but less power than the unsplit version of MLW. The multiple landmarks approach MMLW, which uses split train and test, is illustrated in Figure 4 in the appendix. For one mode, MMLW's power is lower than MLW-split, but is higher than MMD. However, for two modes, MMLW outperforms MLW. MMLW has the highest power for $q = 3$. Both MMLW-knee and MMLW-ME with maximum $q = 20$ has power close to MLW-split when there is one mode and higher power when there is two modes. To further investigate automatic selection, we generalize the problem to have multiple outlier modes at $\pm 4, \pm 8, \pm 12$. The results in Figure 5 and Figure 6 in the appendix show that both automatic selection methods select a number of landmarks that correlates with the true number of outlier nodes.

## 4.2 MNIST and CIFAR-10

Following prior work (Rabanser et al., 2019; Wang et al., 2022), we create two sample testing scenarios using images from either MNIST or CIFAR-10 (Krizhevsky, 2009) datasets. The first sample $\hat{\mu}$ is drawn uniformly from the training set across the $C = 10$ classes, while the second sample $\hat{\nu}$, representing the shifted distribution, is obtained via non-uniform sampling across the classes in the test set. Specifically, we consider either over- or under-representation of one class to create the distribution shift. We use test three learning representations: first, the original vectorized image representations; second, the vectors of

class probability estimates from the softmax outputs of a pre-trained ResNet classifier as in black box shift detection (BBSD) (Lipton et al., 2018) (the network uses the entire training set, which is generally more than the sample size $m$, and the validation set used for epoch selection); and third, the embeddings from the CLIP vision model `openai/clip-vit-large-patch14-336` (Radford et al., 2021). We compare MLW's power to a variety of baseline methods: The kernel projected Wasserstein distance (KPW) is a generalization of KMS to a vector-valued RKHS (Wang et al., 2022). MMD-NTK is a test that uses a neural network trained as the unnormalized witness function for MMD (Cheng & Xie, 2021). MMD-D optimizes a two-layer neural network with 50 hidden units and 50 output units and log-space optimized Gaussian kernel bandwidths creating a joint kernel on the network's output and the original representation (Liu et al., 2020). The mean-embedding (ME) approach (Jitkrittum et al., 2016) optimizes a kernel-based score function (Chwialkowski et al., 2015) in terms of both the Gaussian kernel bandwidth and the locations of multiple landmark points (not constrained to be points in the samples). ME1, as described above, selects a single landmark from the samples. Projected Wasserstein (PW) (Wang et al., 2021) maximizes the Wasserstein distance in terms of a 3-dimensional projection. Training and test splits were 50%-50%.

### 4.2.1 Detecting and localizing over-representation for one class in MNIST (Experiment 1)

Following the MNIST benchmark in Wang et al. (2022), one digit is over-represented: $\mu$ is uniform across all digits and $\nu$ has an **increased** prevalence of one digit. Specifically, $\nu$ is a mixture distribution comprising 85% uniform sampling across classes and 15% probability mass assigned to the designated digit. The discrepancy in prevalence for the majority class is then $|0.15 \cdot 1.0 + 0.85 \cdot 0.1 - 0.1| = 0.135$. Ten trials of the power test were performed. In each trial, 100 Monte Carlo samples were used to compute power, with $B = 100$ permutations. The mean and standard-deviation of the statistical power are reported in Table 2 for sample sizes of 200 and 500.[3] For each class the none-split methods (MMD, MLW, and ME1) have superior statistical power. At

Table 2: Test power (average and standard deviation across 10 runs) on MNIST when upsampling one class/digit. Best method and ties (at 2 decimal digits) are boldfaced, second best are underlined, and third best are italicized.

| | MMD-NTK | MMD-D | ME | PW | KPW | ME1-split | MLW-split | MMD | ME1 | MLW |
|---|---|---|---|---|---|---|---|---|---|---|
| | | | | | **n = 200** | | | | | |
| 0 | 0.801±0.050 | 0.726±0.046 | 0.328±0.140 | 0.324±0.187 | 0.319±0.198 | 0.695±0.114 | *0.832±0.056* | 0.984±0.020 | 0.977±0.028 | **0.995±0.012** |
| 1 | 0.639±0.172 | 0.696±0.079 | 0.307±0.167 | 0.302±0.184 | 0.575±0.219 | 0.439±0.265 | *0.775±0.232* | 0.945±0.093 | *0.780±0.204* | **0.987±0.033** |
| 2 | *0.564±0.191* | 0.431±0.043 | 0.263±0.120 | 0.163±0.093 | 0.183±0.142 | 0.378±0.221 | 0.379±0.215 | **0.750±0.305** | **0.754±0.294** | 0.673±0.284 |
| 3 | 0.463±0.209 | 0.430±0.034 | 0.150±0.084 | 0.151±0.083 | 0.179±0.111 | 0.421±0.258 | 0.335±0.309 | **0.773±0.166** | 0.738±0.185 | *0.615±0.259* |
| 4 | 0.484±0.074 | 0.385±0.031 | 0.146±0.057 | 0.172±0.100 | 0.176±0.096 | 0.353±0.192 | 0.324±0.219 | 0.668±0.227 | **0.729±0.233** | *0.606±0.189* |
| 5 | 0.222±0.135 | *0.227±0.034* | 0.112±0.029 | 0.113±0.051 | 0.101±0.069 | 0.141±0.111 | 0.187±0.166 | 0.331±0.201 | **0.336±0.194** | 0.249±0.130 |
| 6 | *0.710±0.094* | 0.565±0.044 | 0.270±0.108 | 0.199±0.131 | 0.194±0.172 | 0.468±0.256 | 0.607±0.283 | 0.897±0.125 | 0.891±0.118 | **0.947±0.076** |
| 7 | *0.716±0.107* | 0.513±0.036 | 0.143±0.065 | 0.187±0.079 | 0.211±0.189 | 0.313±0.237 | 0.503±0.264 | 0.843±0.146 | 0.713±0.251 | **0.859±0.122** |
| 8 | 0.241±0.124 | 0.303±0.035 | 0.131±0.055 | 0.086±0.039 | 0.092±0.031 | 0.317±0.164 | 0.146±0.126 | 0.593±0.301 | **0.618±0.307** | *0.399±0.275* |
| 9 | 0.442±0.064 | 0.381±0.046 | 0.200±0.079 | 0.102±0.043 | 0.211±0.165 | 0.402±0.148 | 0.297±0.246 | **0.680±0.218** | 0.659±0.258 | 0.660±0.150 |
| | 0.528 | 0.466 | 0.205 | 0.180 | 0.224 | 0.393 | 0.439 | **0.746** | 0.720 | *0.699* |
| | | | | | **n = 500** | | | | | |
| 0 | *0.984±0.016* | *0.982±0.012* | 0.751±0.211 | 0.542±0.234 | 0.527±0.396 | 0.990±0.012 | 0.993±0.013 | **1.000±0.000** | **1.000±0.000** | **1.000±0.000** |
| 1 | 0.842±0.288 | 0.977±0.013 | 0.793±0.320 | 0.603±0.173 | *0.929±0.160* | 0.906±0.208 | **0.996±0.007** | **1.000±0.000** | 0.980±0.057 | **1.000±0.000** |
| 2 | 0.864±0.103 | *0.883±0.017* | 0.514±0.226 | 0.287±0.192 | 0.150±0.166 | 0.865±0.137 | 0.933±0.128 | **1.000±0.000** | **1.000±0.000** | **1.000±0.000** |
| 3 | 0.826±0.168 | *0.889±0.027* | 0.436±0.216 | 0.264±0.192 | 0.186±0.158 | 0.888±0.105 | 0.881±0.258 | **1.000±0.000** | **1.000±0.000** | **1.000±0.000** |
| 4 | 0.810±0.136 | 0.837±0.031 | 0.401±0.187 | 0.311±0.153 | 0.225±0.171 | 0.776±0.246 | *0.866±0.253* | **0.998±0.004** | **0.997±0.005** | 0.990±0.015 |
| 5 | 0.605±0.194 | 0.633±0.040 | 0.270±0.092 | 0.189±0.094 | 0.168±0.101 | 0.584±0.176 | 0.550±0.333 | **0.965±0.045** | 0.936±0.059 | *0.914±0.061* |
| 6 | 0.971±0.034 | *0.956±0.019* | 0.575±0.248 | 0.416±0.286 | 0.437±0.295 | 0.903±0.226 | **0.998±0.004** | **1.000±0.000** | **1.000±0.000** | **1.000±0.000** |
| 7 | 0.959±0.052 | *0.921±0.026* | 0.653±0.298 | 0.455±0.207 | 0.362±0.322 | 0.850±0.218 | 0.894±0.266 | **1.000±0.000** | **0.997±0.009** | **1.000±0.000** |
| 8 | 0.583±0.177 | 0.760±0.044 | 0.312±0.213 | 0.215±0.143 | 0.174±0.196 | 0.725±0.310 | *0.853±0.255* | **0.997±0.006** | **0.999±0.003** | 0.984±0.022 |
| 9 | 0.779±0.157 | *0.823±0.037* | 0.514±0.262 | 0.272±0.133 | 0.306±0.185 | 0.771±0.240 | 0.862±0.253 | **0.995±0.008** | *0.976±0.059* | 0.986±0.033 |
| | 0.822 | 0.866 | 0.522 | 0.355 | 0.346 | 0.826 | *0.883* | **0.995** | 0.989 | 0.987 |

$n = 200$, MMD performs best for 3 of the 10 digits (2nd best for the remainder), ME1 performs best for 4 digits (second best for 3, and third best for 1), and MLW performs best for 4 digits (second best for 3, and third best for 3). MLW-split outperforms ME1-split, and all other split baselines at $n = 500$ (at $n = 200$ MMD-NTK and MMD-D outperform it). We also ran MMLW with knee selection (data not shown). It had slightly lower performance than MLW-split with a power of 0.424 for $n = 200$ and 0.842 for $n = 500$.

---

[3]Results for digit 1 across a wider range of samples size were reported by Wang et al. (2022), as shown in Tables 8 and 9. To study the effect of kernel choice and $p \in \{1, 2\}$, we repeat the experiment for MLW and MMD with different kernels, considering the Laplacian kernel $\kappa(x, y) = e^{-\frac{\|x-y\|}{\sigma}}$ and the triangular kernel $\kappa(x, y) = \max\{0, 1 - \frac{1}{2\sigma}\|x - y\|\}$, where $\sigma$ is the median distance. Results in Table 10 in the appendix show that MMD with Laplacian kernel performs the best on average; and for MLW, the Gaussian kernel with $p = 2$ (default settings) is best.

Representative run times are in Table 3. While MLW (using the full samples) is more computationally intensive than MMD and ME1 (107 s compared to 12–13 s for $n = 2000$), the split version is much faster than any other method. In particular, the kernel max-sliced Wasserstein distance (in the form of KPW) took 4000 s for $n = 2000$. For the baselines that optimize neural networks, MMD-NTK which is linear in sample size is much faster than MMD-D, which requires evaluating the MMD test statistic, while having similar performance. MLW-split is the fastest due to its simple training and linear time testing. The run time for MMLW is slightly higher with $n = 2000$ taking 0.935 s.

Table 3: Representative run time (s) in Google Colab CPU (MMD-D* uses T4 GPU) environment across methods for different sample sizes $n$ for MNIST using original representation (dimension 784).

| $n$ | MMD-NTK | MMD-D | MMD-D* | ME | PW | KPW | ME1-split | MLW-split | MMD | ME1 | MLW |
|---|---|---|---|---|---|---|---|---|---|---|---|
| 500 | 0.299 | 13.16 | 7.98 | 4.38 | 13.21 | 63.02 | 0.079 | **0.050** | 0.481 | 0.962 | 2.53 |
| 1000 | 0.667 | 42.72 | 8.10 | 5.72 | 29.81 | 454.64 | **0.179** | 0.621 | 2.040 | 4.990 | 8.12 |
| 2000 | 1.190 | 164.20 | 10.74 | 7.55 | 107.40 | 4049.35 | 0.680 | **0.495** | 12.20 | 13.07 | 28.32 |

The accuracy of the correspondence of the landmark to the upsampled class is reported in Table 4. MLW consistently outperforms at MMD and ME1 baselines at $n = 500$.

Table 4: Landmark accuracy (average and standard deviation across 10 runs) on MNIST when upsampling one class/digit. Best method and ties are boldfaced.

| $n$ | | Digit 0 | Digit 1 | Digit 2 | Digit 3 | Digit 4 | Digit 5 | Digit 6 | Digit 7 | Digit 8 | Digit 9 | Ave. |
|---|---|---|---|---|---|---|---|---|---|---|---|---|
| 200 | MMD | 0.30±0.46 | 0.20±0.40 | 0.00±0.00 | 0.20±0.40 | 0.00±0.00 | 0.00±0.00 | 0.00±0.00 | 0.10±0.30 | 0.30±0.46 | 0.30±0.46 | 0.14 |
| | ME1 | 0.90±0.30 | 0.60±0.49 | **1.00±0.00** | 0.90±0.30 | **1.00±0.00** | 0.60±0.49 | 0.90±0.30 | 0.60±0.49 | **0.80±0.40** | **0.70±0.46** | 0.80 |
| | MLW | **1.00±0.00** | **1.00±0.00** | 0.90±0.30 | **1.00±0.00** | 0.90±0.30 | 0.60±0.49 | 0.90±0.30 | **0.80±0.40** | 0.70±0.46 | **0.70±0.46** | **0.85** |
| 500 | MMD | 0.10±0.30 | 0.20±0.40 | 0.10±0.30 | 0.00±0.00 | 0.00±0.00 | 0.10±0.30 | 0.00±0.00 | 0.00±0.00 | 0.10±0.30 | 0.30±0.46 | 0.09 |
| | ME1 | **1.00±0.00** | **1.00±0.00** | 0.80±0.40 | 0.80±0.40 | 0.80±0.40 | 0.30±0.46 | 0.90±0.30 | 0.90±0.30 | **1.00±0.00** | 0.80±0.40 | 0.83 |
| | MLW | **1.00±0.00** | **1.00±0.00** | **1.00±0.00** | **1.00±0.00** | **1.00±0.00** | **0.80±0.40** | **1.00±0.00** | **1.00±0.00** | **1.00±0.00** | **1.00±0.00** | **0.98** |

### 4.2.2 Detecting and Localizing Under-Representation (Experiment 2)

Following Rabanser et al. (2019) we consider detecting under-representation of one class (knock-out of 50%) on both MNIST and CIFAR-10 datasets using various learning representations. The minority class has prevalence 0.05 in the test set (remaining classes have prevalence 0.105̄) while the training set has uniform class distribution 0.1 prevalence. Note that the magnitude of the maximum discrepancy in prevalence is 0.05, which is smaller than in Experiment 1 making this a more challenging detection. Based on the results in Experiment 1 we only consider MMD, ME1, and MLW without splitting. Table 5 details the power at sample sizes of 500, 1000, and 2000. Per class power results are in Tables 11 and 12 in the appendix.

Table 5: Average statistical power ($\alpha = 0.05$) on MNIST and CIFAR-10 across representations for detecting imbalanced distributions when the test set has a minority class with a prevalence of 5% corresponding to a drop rate of $p = 50\%$. Power is calculated across 100 random draws and tests use 100 random shuffling to generate the surrogate null distribution. Values are average and standard deviation of power across the 10 classes. The best performance per learning representation and sample size is bolded.

| | | MNIST | | | CIFAR-10 | | |
|---|---|---|---|---|---|---|---|
| | $n$ | MMD | ME1 | MLW | MMD | ME1 | MLW |
| Orig | 500 | **0.292 ± 0.169** | 0.244 ± 0.107 | 0.183 ± 0.218 | 0.043 ± 0.023 | **0.073 ± 0.025** | 0.062 ± 0.030 |
| | 1000 | **0.570 ± 0.161** | 0.482 ± 0.180 | 0.324 ± 0.285 | 0.063 ± 0.031 | 0.077 ± 0.020 | **0.086 ± 0.032** |
| | 2000 | **0.935 ± 0.054** | 0.857 ± 0.130 | 0.609 ± 0.247 | **0.161 ± 0.086** | 0.147 ± 0.051 | 0.149 ± 0.066 |
| CLIP | 500 | **0.221 ± 0.056** | 0.180 ± 0.048 | 0.220 ± 0.160 | 0.214 ± 0.052 | 0.216 ± 0.081 | **0.246 ± 0.114** |
| | 1000 | **0.494 ± 0.148** | 0.377 ± 0.131 | 0.388 ± 0.237 | 0.399 ± 0.125 | 0.379 ± 0.153 | **0.459 ± 0.200** |
| | 2000 | **0.874 ± 0.132** | 0.747 ± 0.189 | 0.699 ± 0.264 | 0.758 ± 0.091 | 0.724 ± 0.162 | **0.813 ± 0.197** |
| BBSD | 500 | 0.474 ± 0.074 | **0.551 ± 0.081** | 0.400 ± 0.109 | 0.253 ± 0.021 | **0.343 ± 0.057** | 0.311 ± 0.044 |
| | 1000 | 0.164 ± 0.085 | **0.894 ± 0.039** | 0.777 ± 0.080 | 0.281 ± 0.066 | **0.691 ± 0.069** | 0.632 ± 0.106 |
| | 2000 | 0.681 ± 0.120 | **0.999 ± 0.003** | 0.997 ± 0.005 | 0.607 ± 0.084 | **0.959 ± 0.034** | 0.948 ± 0.049 |

The best performing method depends on the representation and dataset. The original vectorized image representations yields meaningful detection performance on MNIST due to the prototypical nature of digits,

but yields much lower statistical power for CIFAR-10. Using CLIP offers better power for both methods and both datasets, with MMD performing best for MNIST and MLW performing best for CIFAR-10. For BBSD, which gives the highest performance overall, ME1 performs best on all sample sizes for both methods, with MLW second best on a majority of cases.

We now show how the witness function, defined by the kernel landmark, is reliably interpretable as an indicator of the specific discrepancies associated with the class imbalance. Namely, we examine the landmark for MLW (or ME1) and the point with the largest magnitude witness function evaluations for MMD. Ideally, these points should be from the minority class. Examples of these instances for the CIFAR-10 dataset using the BBSD and CLIP learning representation are shown in Figure 2, a set of 10 runs is found in Figure 9 along with the MNIST results in Figure 10 in the appendix. MLW with BBSD consistently selects landmarks from the minority class. In contrast, the images with the largest MMD witness function evaluations often do not correspond to the localized class imbalance.

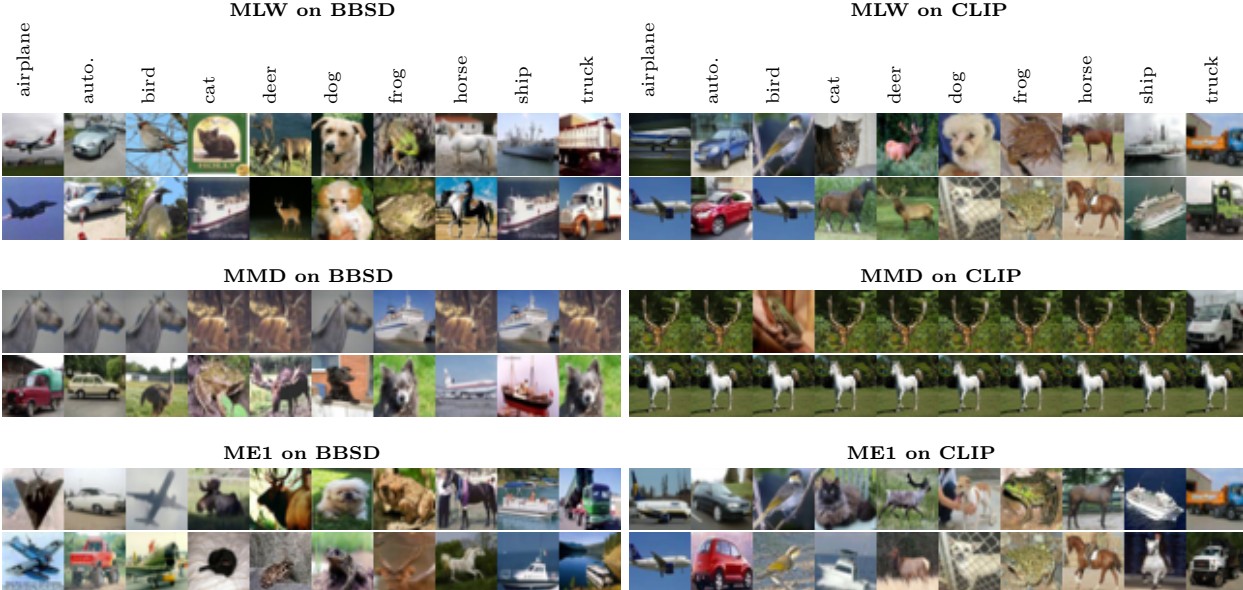

Figure 2: Landmarks chosen for CIFAR-10 when one class (indicated by column header) is a minority in the test set while the training set has uniform class distribution. Sample size is $m = n = 2000$. Images in each row correspond to different trials with each column uses the same training set, only the test set is modified.

Table 6 shows the average landmark accuracy for MLW is higher than ME1 or MMD for larger sample sizes. Generally, accuracy on CLIP is higher than the original representation, and the BBSD representations has the highest accuracy. Table 13 and Table 14 in the appendix detail the landmark accuracy per class.

### 4.3 Higgs Dataset (Experiment 3)

To test on a dataset without localized shift, we apply MLW and MMLW to the Higgs dataset introduced by Chwialkowski et al. (2015) and derived from high-energy physics data (Whiteson, 2014; Baldi et al., 2014). This is a challenging test and larger sample sizes (up to 10,000 points in training from each distribution) are used. Details and comparison of power results to published baselines are in Appendix A.5.5 and Table 16. The results show that while MLW is not competitive with methods that learn deep feature representations or classifiers (but can be used in conjunction), MMLW outperforms the ME test, which is the most similar in design, for larger sample sizes when using a similar number of inducing points to the number of landmarks. The ME test statistic with the MMLW selected landmarks outperforms or matches the best baselines at multiple sizes above 3000. Results comparing MLW, MMLW, ME, and ME1 are in Table 7. Regarding MMLW, the MMLW-ME that uses the estimate of test power to select the number has the best performance

Table 6: Accuracy of the landmark corresponding to knockout distribution shift on CIFAR-10. After knock-out the minority class has a prevalence of 5% compared to the other classes having 10.05%. Accuracy is calculated across 100 random draws and averaged across classes.

| Dataset | Method | $m = n = 500$ | | | $m = n = 1000$ | | | $m = n = 2000$ | | |
| --- | --- | --- | --- | --- | --- | --- | --- | --- | --- | --- |
| | | Orig | CLIP | BBSD | Orig | CLIP | BBSD | Orig | CLIP | BBSD |
| MNIST | MMD | 0.113 | 0.117 | 0.130 | 0.109 | 0.091 | 0.103 | 0.107 | 0.113 | 0.111 |
| | ME1 | 0.142 | 0.305 | **0.772** | 0.161 | 0.426 | 0.891 | 0.174 | 0.538 | 0.880 |
| | MLW | **0.281** | **0.371** | 0.732 | **0.361** | **0.573** | **0.952** | **0.468** | **0.742** | **0.998** |
| CIFAR-10 | MMD | **0.098** | 0.110 | 0.170 | **0.099** | 0.123 | 0.183 | 0.103 | 0.111 | 0.190 |
| | ME1 | 0.069 | **0.464** | 0.279 | 0.092 | 0.659 | 0.370 | 0.071 | 0.872 | 0.406 |
| | MLW | 0.080 | 0.403 | **0.621** | 0.095 | **0.667** | **0.860** | **0.112** | **0.895** | **0.979** |

Table 7: Power ($\alpha = 0.05$) for MLW-split, MMLW ($q = 13$, with knee, or with ME), ME (original or using MMLW landmarks), and ME1-split on the Higgs dataset. Entries are average±standard error across 5 runs. Column headings are $n^{\text{Tr}} = n^{\text{Te}}$.

| Method | 1000 | 2000 | 3000 | 4000 | 5000 | 6000 | 8000 | 10000 |
| --- | --- | --- | --- | --- | --- | --- | --- | --- |
| MLW-split | 0.048±0.007 | 0.086±0.010 | 0.150±0.017 | 0.206±0.010 | 0.300±0.011 | 0.438±0.023 | 0.612±0.022 | 0.764±0.022 |
| MMLW | 0.068±0.012 | 0.146±0.011 | 0.202±0.014 | 0.302±0.019 | 0.376±0.011 | 0.522±0.017 | 0.726±0.012 | 0.876±0.014 |
| w/ knee | 0.054±0.011 | 0.124±0.008 | 0.168±0.014 | 0.238±0.011 | 0.344±0.012 | 0.494±0.019 | 0.700±0.005 | 0.834±0.014 |
| w/ ME | 0.070±0.011 | 0.136±0.011 | 0.212±0.007 | 0.340±0.017 | 0.428±0.012 | 0.606±0.021 | 0.800±0.015 | 0.912±0.012 |
| ME | **0.120±0.007** | 0.165±0.019 | 0.197±0.012 | - | 0.410±0.041 | - | 0.691±0.067 | 0.786±0.041 |
| w/MMLW | 0.106±0.011 | **0.234±0.025** | **0.422±0.017** | **0.604±0.028** | **0.718±0.020** | **0.856±0.017** | **0.950±0.004** | **0.986±0.006** |
| ME1-split | 0.034±0.010 | 0.050±0.021 | 0.052±0.019 | 0.058±0.007 | 0.052±0.020 | 0.072±0.029 | 0.072±0.041 | 0.072±0.040 |

on the majority of the sample sizes, outperforming fixed $q = 13 \approx 2000^{1/3}$ and $q = 1$ (MLW) as well as outperforming the MMLW-knee heuristic. While MMLW-knee has higher power than a single landmark, but has less than always using 13 landmarks. The distribution of number of landmarks for sample size is shown in Figure 11 in the appendix. The results show that the automatic selection with default Kneedle parameters often selects a single landmark or 19 landmarks, while MMLW-ME selects ar range from 8–18 concentrated around 13 or 14.

On these large sample sizes, MLW and MMLW have relatively fast training. We note the entire landmark selection and permutation test for MLW takes around 30 s at $n = 8000$ on both an Apple M4 Pro with 24 GB RAM and a Python 3 Google Compute Engine backend with 12.7 GB RAM. In comparison, running MMD-D on a GPU (NVIDIA Tesla P100) takes 145 s for $n = 6000$. With an approximate median and batch implementation (see Appendix A.5.5), MMLW's training takes 33.54 s for $m = n = 10000$ and test takes an additional 10 s. Example run times shown in Table 17 in the appendix. For comparison, the state-of-the-art AutoML (Kübler et al., 2022) reports results with run time for training capped to 1, 5, or 10 minutes. Thus, MMLW has the potential to scale to provide statistically powerful and interpretable two-sample tests.

## 5 Conclusion

We introduce the max-sliced kernel landmark Wasserstein distance (MLW) as a divergence measure that can detect and localize shifts in distributions. The method evaluates the discrepancy between two distributions in terms of the similarity to a landmark point, which for a sample can be computed exactly and efficiently. Through statistical power tests, we show that it is competitive with MMD and using only a single point in the mean embedding (ME) test in detecting both imbalanced and perturbed distributions on diverse learning representations, but MLW's landmarks more accurately related to the discrepancy. To investigate multiple discrepancies, the multiple landmarks version of MLW (MMLW) can be applied. Then the ME power estimate can be used to select an appropriate number of landmarks, and the ME test can be applied with the MMLW selected landmarks—yielding power competitive with the state-of-the-art on the Higgs dataset. In conclusion, MLW and MMLW are useful tools to identify localized discrepancies.

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

## A  Appendix

### A.1  Proofs

*Proof of Theorem 1.* To be a probability distance metric for $\mu, \nu, \xi \in P(\mathcal{X})$ with finite $p$th moments, $MLW_{\kappa,p}$ must satisfy the following (Mueller, 1997):

- $MLW_{\kappa,p}(\mu, \nu) \geq 0$

- $MLW_{\kappa,p}(\mu, \nu) = MLW_{\kappa,p}(\nu, \mu)$

- $MLW_{\kappa,p}(\mu, \nu) = 0 \iff \mu = \nu$.

- $MLW_{\kappa,p}(\mu, \nu) \leq MLW_{\kappa,p}(\mu, \xi) + MLW_{\kappa,p}(\nu, \xi)$.

Non-negativity and symmetry are obvious from the proprieties of the Wasserstein distance. If $\mu = \nu$, then for any $\omega \in \mathcal{H}$, $\inf_{\gamma \in \Gamma(\mu, \mu)} \mathbb{E}_{(X,Y) \sim \gamma} |\omega(X) - \omega(Y)|^p = 0$.

We assume $\mu \neq \nu$ and proceed to lower bound the distance and show that $MLW_{\kappa, p}(\mu, \nu) > 0$. For any $\omega \in \mathcal{H}$,

$$W^p_{\mathfrak{d}^p_\omega}(\mu, \nu) = \inf_{\gamma \in \Gamma(\mu, \nu)} \mathbb{E}_{(X,Y) \sim \gamma} |\omega(X) - \omega(Y)|^p$$

$$\geq \inf_{\gamma \in \Gamma(\mu, \nu)} |\mathbb{E}_{(X,Y) \sim \gamma} [\omega(X) - \omega(Y)]|^p \tag{7}$$

$$= |\langle \bar{h}(\mu) - \bar{h}(\nu), \omega \rangle|^p, \tag{8}$$

where the inequality follows from Jensen's inequality based on the convexity of $|\cdot|^p$, and the subsequent equality is based on the definition of mean embedding operator $\bar{h}$. Taking the supremum over the set of landmarks yields an expression in terms of the difference of the means in the RKHS

$$\sup_{\omega \in \mathcal{L}} W^p_{\mathfrak{d}^p_\omega}(\mu, \nu) \geq \sup_{\omega \in \mathcal{L}} |\langle \underbrace{\bar{h}(\mu) - \bar{h}(\nu)}_{h_\Delta \in \mathcal{H}}, \omega \rangle|^p \tag{9}$$

$$= \sup_{z \in \mathcal{X}} |\langle h_\Delta, \phi(z) \rangle|^p = \sup_{z \in \mathcal{X}} |h_\Delta(z)|^p. \tag{10}$$

When $\kappa$ is characteristic, the mean embedding operator $\bar{h}(\zeta) : \zeta \mapsto \mathbb{E}_{X \sim \zeta}[\phi(X)]$ is injective for $\zeta \in \mathcal{P}_\mathcal{X}$ (Fukumizu et al., 2008; Gretton et al., 2012a), and $\mu \neq \nu \implies \bar{h}(\mu) \neq \bar{h}(\nu) \implies \exists f \in \mathcal{F}, \langle h_\Delta, f \rangle \neq 0$. Equivalently, $\mu \neq \nu \implies \|h_\Delta\|_\infty = \sup_{z \in \mathcal{X}} |h_\Delta(z)| > 0$. Together this yields $MLW_{\kappa, p}(\mu, \nu) \geq \sup_{\omega \in \mathcal{L}} |\langle h_\Delta, \omega \rangle|^p > 0$ for $\mu \neq \nu$.

The triangle inequality follows from the fact that the Wasserstein distance $W_{\mathfrak{d}^p_E}$ is itself is a metric:

$$MLW_{\kappa, p}(\mu, \nu) = \sup_{\omega \in \mathcal{L}} W_{\mathfrak{d}^p_E}(\omega \sharp \mu, \omega \sharp \nu)$$

$$\leq \sup_{\omega \in \mathcal{L}} \left( W_{\mathfrak{d}^p_E}(\omega \sharp \mu, \omega \sharp \zeta) + W_{\mathfrak{d}^p_E}(\omega \sharp \nu, \omega \sharp \zeta) \right) \tag{11}$$

$$\leq \sup_{\omega \in \mathcal{L}} W_{\mathfrak{d}^p_E}(\omega \sharp \mu, \omega \sharp \zeta) + \sup_{\omega' \in \mathcal{L}} W_{\mathfrak{d}^p_E}(\omega' \sharp \nu, \omega' \sharp \zeta) \tag{12}$$

$$= MLW_{\kappa, p}(\mu, \zeta) + MLW_{\kappa, p}(\nu, \zeta) \tag{13}$$

$$\square$$

*Proof of Theorem 2.* Define $\mathcal{X}_\mathcal{S} = \{x \in \mathcal{X} : \phi(x) \in \mathcal{S}\} = \mathcal{X} \cap \mathcal{S}$, where $\mathcal{S} = \text{span}(\{\phi(z_i)\}_{i=1}^l)$. For any $z \in \mathcal{X}$, the point's embedding can be decomposed as $\phi(z) = \omega_\mathcal{S} + h$, where $\omega_\mathcal{S} \in \mathcal{S}$ and $h \in \mathcal{H}$ with $\langle h, \phi(z_i) \rangle = h(z_i) = 0, \quad i \in [l]$ such that $\kappa(z_i, z) = \omega_\mathcal{S}(z_i) + h(z_i) = \omega_\mathcal{S}(z_i), \quad i \in [l]$. It follows that $\sup_{z \in \mathcal{X}_\mathcal{S}} W^p_{\mathfrak{d}^p_{\phi(z)}}(\hat{\mu}, \hat{\nu}) = \sup_{z \in \mathcal{X}} W^p_{\mathfrak{d}^p_{\phi(z)}}(\hat{\mu}, \hat{\nu})$. To complete the proof, we need to show that $\mathcal{X}_\mathcal{S} = \mathcal{Z}$. We assume without loss of generality that the points in $\mathcal{Z}$ are distinct. Let $z' \in \mathcal{X}_\mathcal{S}$, then by the definition of $\mathcal{S}$ there exists coefficients $\alpha_1, \dots, \alpha_l$ such that $\phi(z') = \sum_{i=1}^l \alpha_i \phi(z_i)$, which implies that $\phi(z') - \sum_{i=1}^l \alpha_i \phi(z_i) = 0$. Assume by way of contradiction that $z' \notin \mathcal{Z}$, i.e., $\{z'\} \cup \mathcal{Z}$ is a set of $l+1$ distinct points. Since the kernel is strictly positive definite, then $\phi(z'), \phi(z_1), \dots, \phi(z_l)$ are linearly independent, but this contradicts the fact that there are a set of non-zero coefficients $1, -\alpha_1, \dots, -\alpha_l$ such that the combination $1\phi(z') - \sum_l \alpha_l \phi(z_l) = 0$. Therefore, our assumption must be false and $z' \in \mathcal{Z}$. Combined with $\mathcal{Z} \subseteq \mathcal{X}_\mathcal{S}$ this shows that $\mathcal{Z} = \mathcal{X}_\mathcal{S}$. $\square$

## A.2 MLW Computation for General Masses

For general masses or sample sizes, the optimization can be expressed in terms of the non-zero entries of the solution $\acute{\mathbf{P}}^k = \arg\min_{\mathbf{P} \in \mathcal{P}_{\acute{\mu}^k, \acute{\nu}^k}} P_{ij} |i - j|^p$ obtained via the Northwest corner rule for each candidate landmark $k \in [l]$ using the permuted masses $\acute{\boldsymbol{\mu}}^k = [\mu_{\pi_i^k}]_{i=1}^m$ and $\acute{\boldsymbol{\nu}}^k = [\nu_{\sigma_i^k}]_{i=1}^n$,

$$MLW^p_{\kappa, p}(\hat{\mu}, \hat{\nu}) = \max_{k \in [l]} \acute{\mathbf{p}}^{k\top} |(\acute{\mathbf{k}}_{Xz_k})_{\mathbf{i}^k} - (\acute{\mathbf{k}}_{Yz_k})_{\mathbf{j}^k}|^{\circ p}. \tag{14}$$

where $[\acute{p}_t^k]_{t=1}^s = [\acute{P}_{\acute{i}_t\acute{j}_t}]_{t=1}^s$ are the non-zero entries of $\hat{\mathbf{P}}^k$ with vectors of row and column indices $\hat{\mathbf{i}}^k$ and $\hat{\mathbf{j}}^k$ of length-$s$ ($s$, the cardinality of the support of the transport plan is on the order of $n$). Listing 3 provides the code for uniform masses but general sample sizes using the POT (Flamary et al., 2024) toolbox, available as `pip install pot`.

## A.3 Single-sort Permutation Algorithm

For $k \in [m+n]$, let $\boldsymbol{\pi}^k$ denote the permutation for the pooled sample such $\kappa(z_{\pi_1^k}, z_k) \leq \cdots \leq \kappa(z_{\pi_{m+n}^k}, z_k)$. To generate the $b$th surrogate sample, $b \in [B]$, let $\boldsymbol{\xi}^b \in \mathcal{Q}_{m+n}$ denote the permutation vector that selects $m$ indices for the first sample $\mathcal{I}_X^b = \{\xi_i^b\}_{i=1}^m$ and $n$ indices for the second sample $\mathcal{I}_Y^b = \{\xi_{i+m}^b\}_{i=1}^n$. Each column of $\mathbf{K}_{\tilde{X}^b Z} = [\acute{\mathbf{k}}_{\tilde{X}^b z_k}]_{k=1}^{m+n}$ is created as $\acute{\mathbf{k}}_{\tilde{X}^b z_k} = [\kappa(z_{\pi_j^k}, z_k)]_{j \in [m+n]:\pi_j^k \in \mathcal{I}_X^b}$ and the remaining entries form $\mathbf{K}_{\tilde{Y}^b} = [\acute{\mathbf{k}}_{\tilde{Y}^b z_k}]_{k=1}^{m+n}$ with entries $\acute{\mathbf{k}}_{\tilde{Y}^b z_k} = [\kappa(z_{\pi_j^k}, z_k)]_{j \in [m+n]:\pi_j^k \in \mathcal{I}_Y^b}$. Using boolean indexing constructing these matrices require $\mathcal{O}(n^2)$ operations. For $m = n$, the test statistic is

$$\tilde{D}^b = \max_{k \in [2n]} \frac{1}{n} \|\acute{\mathbf{k}}_{\tilde{X}^b z_k} - \acute{\mathbf{k}}_{\tilde{Y}^b z_k}\|_p^p. \tag{15}$$

For general sample sizes, the test statistic is

$$\tilde{D}^b = \max_{k \in [m+n]} \acute{\mathbf{p}}^\top |\mathbf{R}^X \acute{\mathbf{k}}_{\tilde{X}^b z_k} - \mathbf{R}^Y \acute{\mathbf{k}}_{\tilde{Y}^b z_k}|^{\circ p}, \tag{16}$$

where $\mathbf{R}^X \in \{0,1\}^{s \times m}$ and $\mathbf{R}^Y \in \{0,1\}^{s \times n}$ are sparse binary matrices with the $s$ ones at the row-column indices $\{(t, \acute{i}_t)\}_{t=1}^s$ and $\{(t, \acute{j}_t)\}_{t=1}^s$ corresponding to the indexing of rows of the kernel matrices corresponding to $\acute{\mathbf{i}}$ and $\acute{\mathbf{j}}$.

## A.4 Code Listings

Code repository will be made available with publication.

```python
import numpy as np

def ME1(K_XZ,K_YZ):
    m = len(K_XZ)
    n = len(K_YZ)
    assert n==m , 'sample size must be equal'
    deltas = K_XZ - K_YZ
    delta_bars = np.mean(deltas,axis=0)
    delta_vars = np.var(deltas,axis=0)
    test_stats =  float(n)*(delta_bars**2)/delta_vars
    idx = np.argmax(test_stats)
    return idx, test_stats[idx]
```

Listing 1: NumPy implementation of the empirical ME1 test statistic and landmark for two samples of size $m = n$, given precomputed kernel matrices that are each $n \times 2n$.

```python
import numpy as np
from numba import njit, prange

def MLW(K_XZ, K_YZ, p):
    LSKW = np.mean(np.abs(np.sort(K_XZ,axis=0)-np.sort(K_YZ,axis=0))**p,axis=0)**(1/p)
    k_star = np.argmax(LSKW)
    return LSKW[k_star], k_star

def MLW_pvalue(K_XZ,K_YZ,p,permutations):
    div,k_star  = MLW_sorted(np.sort(K_XZ,axis=0),np.sort(K_YZ,axis=0),p)
    K = np.vstack((K_XZ,K_YZ))
    x_idx = np.concatenate((np.ones(len(K_XZ)),np.zeros(len(K_YZ)))).astype(np.bool)
    pi_ks, K_acute = np.argsort(K,axis=0), np.sort(K,axis=0)
    shuffled_tests = permute_MLW_once(K_acute,pi_ks,x_idx,p,permutations)
    p_value = np.mean(div**p <= shuffled_tests)
    return p_value, k_star

@njit(parallel=True, fastmath=True)
def permute_MLW_once(K_acute,pi_ks,x_idx,p,permutations):
    shuffled_tests = np.zeros(permutations)
    for b in prange(permutations):
        x_idx_b = x_idx[np.random.permutation(len(x_idx))]
        per_landmark = 0
        for k in range(len(K_acute)):
            x_idx_k = x_idx_b[pi_ks[:,k]]
            val = np.mean(np.abs(K_acute[x_idx_k==1,k] - K_acute[x_idx_k==0,k])**p)
            per_landmark = np.maximum(per_landmark, val)
        shuffled_tests[b] = per_landmark
    return shuffled_tests
```

Listing 2: NumPy implementation of the empirical LSKW distances, MLW distance, the optimal landmark, and the p-value for the test in the case of $m = n$ given two precomputed kernel matrices that that are each $n \times 2n$.

```python
import numpy as np
import ot

def MLW(K_XZ,K_YZ,p):
    return MLW_sorted(np.sort(K_XZ,axis=0),np.sort(K_YZ,axis=0),p)

def MLW_sorted(K_XZ, K_YZ, p):
    m = K_XZ.shape[0]
    n = K_YZ.shape[0]
    if m==n:
        LSKW = np.mean(np.abs(K_XZ - K_YZ)**p, axis=0)**(1/p)
    else:
        acute_p, acute_i, acute_j = northwest_corner_rule_iid(m,n)
        LSKW = (acute_p[:,np.newaxis].T @ np.abs(K_XZ[acute_i]-K_YZ[acute_j])**p).squeeze()**(1/p)
    k_star = np.argmax(LSKW)
    return LSKW[k_star], k_star

def northwest_corner_rule_iid(m,n):
    acute_p,acute_indices,_ = ot.lp.emd_wrap.emd_1d_sorted(
        np.full(m,1/m),np.full(n,1/n),1.0+np.arange(m),1.0+np.arange(n))
    acute_i, acute_j = acute_indices[:,0],acute_indices[:,1]
    return acute_p, acute_i, acute_j
```

Listing 3: NumPy implementation of the empirical MLW distance and p-value for general sample sizes $m$ and $n$, given precomputed kernel matrices.

```python
import numpy as np
from scipy.spatial.distance import pdist, squareform

def rbf_kernel(X,kernel_name='gauss'):
    n,d = X.shape
    d2 = pdist(X,'sqeuclidean')
    D2 = squareform(d2)
    sigma = np.median(d2)
    if kernel_name=='gauss':
        rbf2 = lambda d2:  np.exp(-d2/(2*sigma**2))
    elif kernel_name =='laplace':
        rbf2 = lambda d2:  np.exp(-np.sqrt(d2)/sigma)
    elif kernel_name =='tri':
        rbf2 = lambda d2:  np.maximum(0,1-np.sqrt(d2)/(2*sigma))
    else:
        rbf2 = kernel_name
    K = rbf2(D2)
    K  = (K+np.transpose(K))/2
    return K,rbf2
```

Listing 4: NumPy/SciPy implementation of radial basis function kernels with median heuristic.

```python
import numpy as np
from kneed import KneeLocator
from scipy import stats

def MMLW_train(X,Y,max_iter=10,p=2,kernel='gauss',select_option=None,alpha=0.05):
    Z = np.vstack((X,Y))
    K, rbf2 = rbf_kernel(Z,kernel)
    m,n = len(X), len(Y)
    x_idx = np.concatenate((np.ones(m), np.zeros(n)))
    K_XZ, K_YZ = K[x_idx==1, :], K[x_idx==0, :]
    LSKW = np.mean(np.abs(np.sort(K_XZ,axis=0) - np.sort(K_YZ,axis=0))**p, axis=0)**(1/p)
    div, c_divs, indices = 0, [], []
    for iter in range(max_iter):
        logdets = -np.inf*np.ones(m+n)
        for i in range(m+n):
            if i not in indices:
                cand_indices =np.array(indices+[i],dtype=int)
                KC = K[cand_indices,:][:,cand_indices]
                _,logdets[i] = np.linalg.slogdet(KC)
        idx = np.argmax(LSKW + logdets)
        indices.append(idx)
        div += LSKW[idx]
        c_divs.append(div)
    if select_option == 'knee':
        kl = KneeLocator(np.arange(max_iter), c_divs,  curve="concave", direction="increasing")
        break_pt = kl.knee+1 if kl.knee==0 else kl.knee
        indices = indices[:break_pt]
    elif select_option == 'ME':
        est_powers = [ estimate_me_power(K_XZ[:,indices[:q+1]], K_YZ[:,indices[:q+1]], alpha) for q in range(max_iter)]
        indices = indices[:np.argmax(est_powers)+1]
    return Z[indices], rbf2

def estimate_me_power(K_XZ, K_YZ, alpha=0.05, return_stat=False):
    deltas = K_XZ - K_YZ
    n, q = deltas.shape
    mu_hat = np.mean(deltas, axis=0)
    sigma_hat = np.cov(deltas, rowvar=False)
    if q>1:
        sigma_hat += 1/q*np.trace(sigma_hat)*0.01 * np.eye(q) # regularization
        try:
            x = np.linalg.solve(sigma_hat, mu_hat)
        except np.linalg.LinAlgError:
            x = np.linalg.lstsq(sigma_hat, mu_hat, rcond=None)[0]
        stat = n * (mu_hat.T @ x)
    else:
        stat = n * (mu_hat**2/sigma_hat)[0]
    return stat if return_stat else stats.ncx2.sf(stats.chi2.ppf(1 - alpha, df=q), df=q, nc=max(0,stat))

def ME_test(X,Y,Z_stars,rbf2, permutations = 100):
    Z = np.vstack((X, Y))
    m,n = len(X),len(Y)
    x_idx = np.concatenate((np.ones(m), np.zeros(n))).astype(bool)
    K_Z = rbf2(np.hstack( [np.sum( np.abs(Z-z_star)**2, axis=1,keepdims=True) for z_star in Z_stars] ))
    stat = estimate_me_power(K_Z[x_idx],K_Z[~x_idx],return_stat=True)
    shuffled_tests = np.zeros(permutations)
    for b in range(permutations):
        x_idx_b = x_idx[np.random.permutation(len(x_idx))]
        shuffled_tests[b] = estimate_me_power(K_Z[x_idx_b],K_Z[~x_idx_b],return_stat=True)
    p_val = np.mean(stat <= shuffled_tests)
    return p_val
```

Listing 5: NumPy implementations of the MMLW selection of landmarks with optional Kneedle method or ME-based landmark cardinality selection and ME test, assuming $m = n$ and vector valued data.

```python
from numba import njit, prange

def MMLW_test(X,Y,Z_stars,rbf2, p=2, permutations = 100):
    Z = np.vstack((X, Y))
    m,n = len(X),len(Y)
    x_idx = np.concatenate((np.ones(m), np.zeros(n)))
    K_Z = rbf2(np.hstack( [np.sum( np.abs(Z-z_star)**2, axis=1,keepdims=True) for z_star in Z_stars] ))
    MMLW = np.sum((np.mean(np.abs(np.sort(K_Z[x_idx==0],axis=0)
                                  - np.sort(K_Z[x_idx==1],axis=0)) ** p,axis=0)**(1/p)))
    shuffled_tests = permute_MMLW_once(np.sort(K_Z,axis=0),np.argsort(K_Z,axis=0),x_idx,p,permutations)
    p_val = np.mean(MMLW <= shuffled_tests)
    return p_val

@njit(parallel=True, fastmath=True)
def permute_MMLW_once(K_acute,pi_ks,x_idx,p,permutations):
    shuffled_tests = np.zeros(permutations)
    for b in prange(permutations):
        x_idx_b = x_idx[np.random.permutation(len(x_idx))]
        per_landmark = 0
        for k in range(K_acute.shape[1]):
            x_idx_k = x_idx_b[pi_ks[:,k]]
            val = np.mean(np.abs(K_acute[x_idx_k==1,k] - K_acute[x_idx_k==0,k])**p)**(1/p)
            per_landmark += val
        shuffled_tests[b] = per_landmark
    return shuffled_tests
```

Listing 6: NumPy implementation of the MMLW test, assuming $m = n$ and vector valued data.

## A.5    Additional Results

### A.5.1    Additional Results for Toy Example

Figure 3 illustrates the operation of the MLW and MMD for 1D toy examples. Figure 4 illustrates the operation of the MMLW for 1D toy examples. The number of landmarks selected by the knee method is shown in Figure 5, and the number of landmarks selected by MMLW-ME is shown in Figure 6.

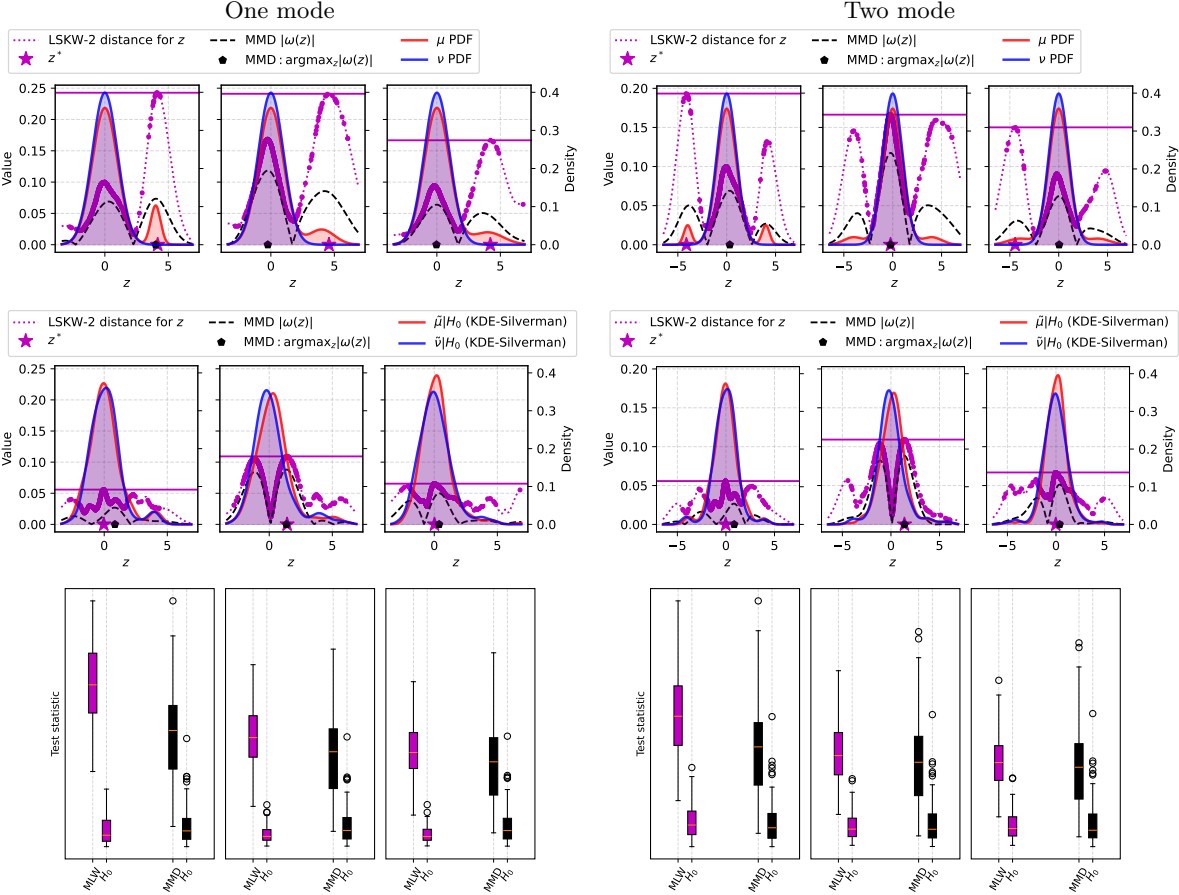

Figure 3: Illustration of MLW and MMD for toy example. (Top) The purple dots and dotted curves show the landmark sliced kernel Wasserstein-2 distance (LSKW-2) as a function of $z \in \mathcal{X} = \mathbb{R}$ (star); the dashed line is the absolute value of the MMD witness function whose maximum is indicated by hexagon; and the solid blue and red lines show the true PDF. (Middle) Results under $H_0$, where a single permutation is used to sample from the surrogate null distribution. The purple dots and dotted curves show the landmark sliced kernel Wasserstein-2 distance (LSKW-2) as a function of $z \in \mathcal{X} = \mathbb{R}$; the dashed line is the absolute value of the MMD witness function; and the solid lines show kernel-density estimates. (Bottom) The box plots show the MLW and MMD test statistics (each on its own scale) for the three cases and under the surrogate null distribution $H_0$.

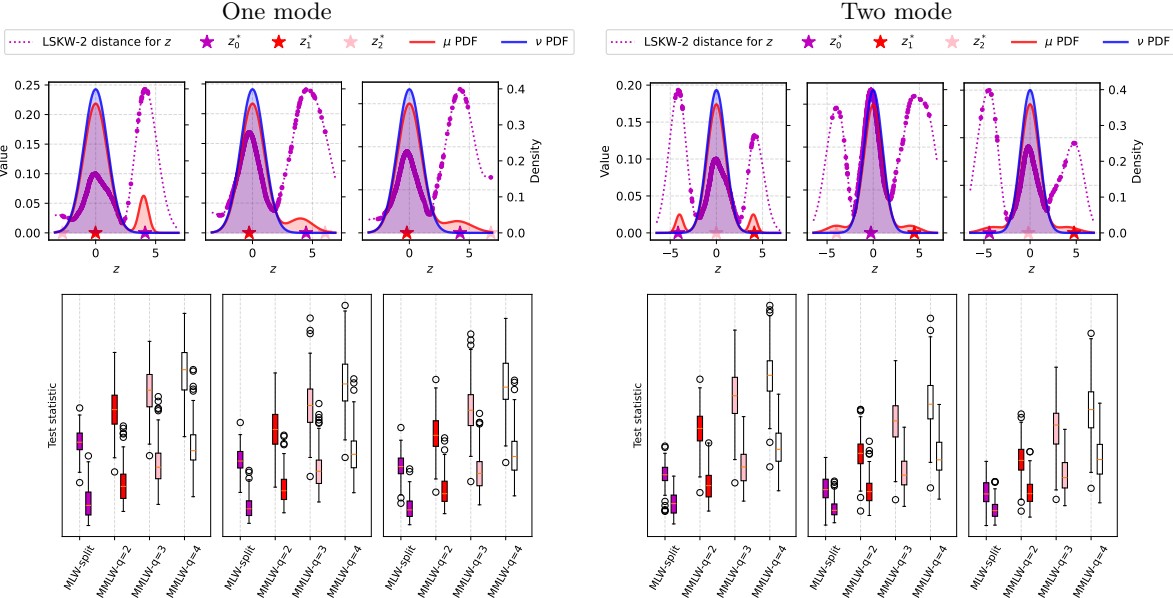

Figure 4: Illustration of MMLW for toy examples when $\mu = \mathcal{N}(0, 1)$ and $\nu = 0.9\mu + 0.1\xi$, where the outlier distribution has either one mode $\xi = \mathcal{N}(4, \sigma^2)$ or two modes $\xi = 0.5\mathcal{N}(-4, \sigma^2) + 0.5\mathcal{N}(4, \sigma^2)$, tested across $\sigma^2 \in \{0.15, 1, 1.15\}$ with $m = n = 250$. (Top) The purple dots and dotted curves show the landmark sliced kernel Wasserstein-2 distance (LSKW-2) as a function of $z \in \mathcal{X} = \mathbb{R}$; the dashed line is the absolute value of the MMD witness function; and the solid lines show the true PDF. (Bottom) The box plots show the MLW-split and MMLW test statistics surrogate null distribution.

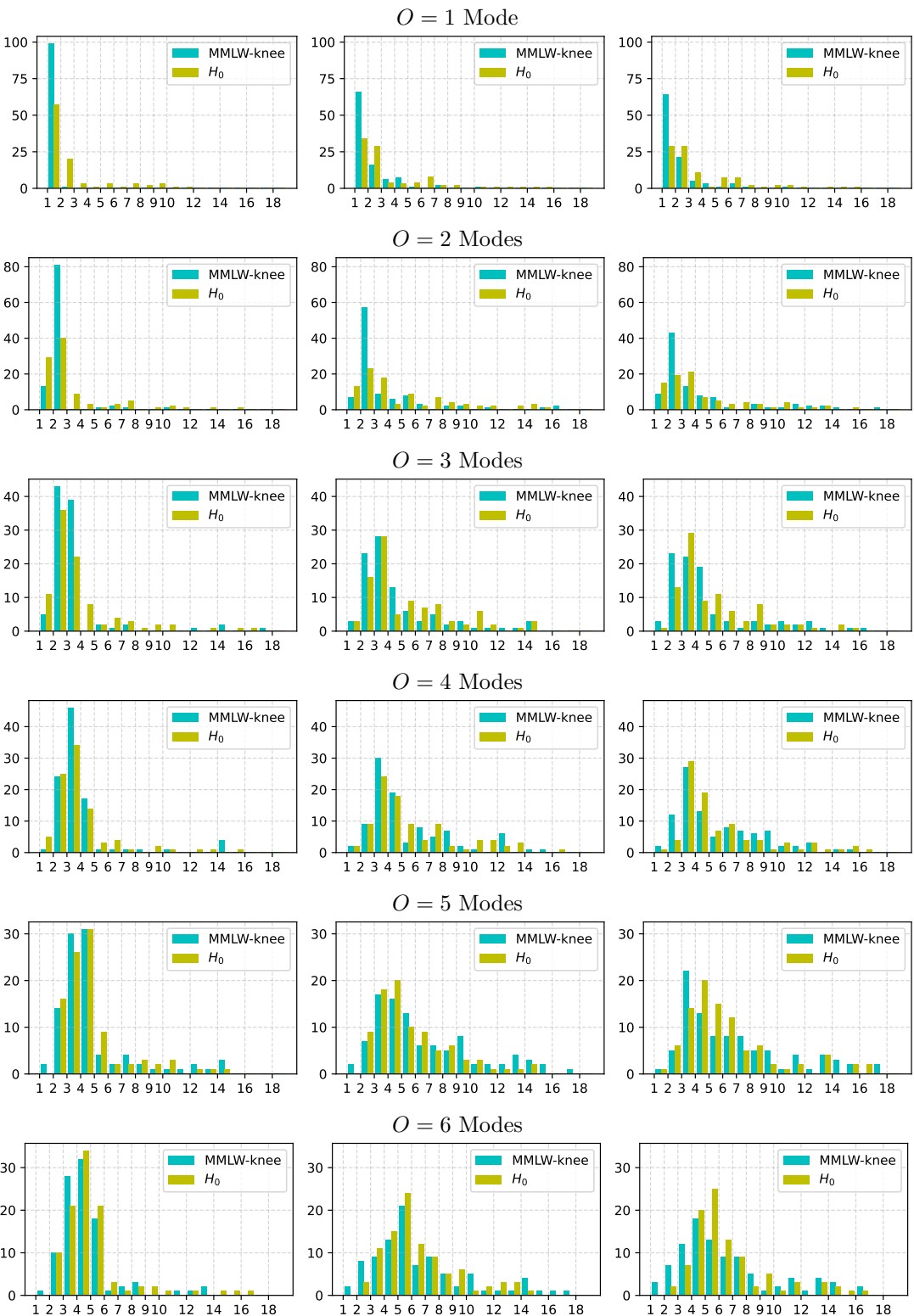

Figure 5: Number of landmarks selected by MMLW-knee method when $\mu = \mathcal{N}(0,1)$ and $\nu = 0.9\mu + 0.1\xi$ (or under $H_0$ where two samples are from $\frac{1}{2}\mu + \frac{1}{2}\nu$), where the outlier distribution $\xi = \frac{1}{O}\sum_{i=1}^{O}\mathcal{N}(a_i, \sigma^2)$ has $O \in \{1, 2, 3, 4, 5, 6\}$ modes at locations in the vector $\mathbf{a} = [-4, 4, -8, 8, -12, 12]$ and variance $\sigma^2 \in \{0.15, 1, 1.15\}$. The landmarks selected under $H_0$ created by a permutation are also shown. Samples sizes are $m = n = 250$.

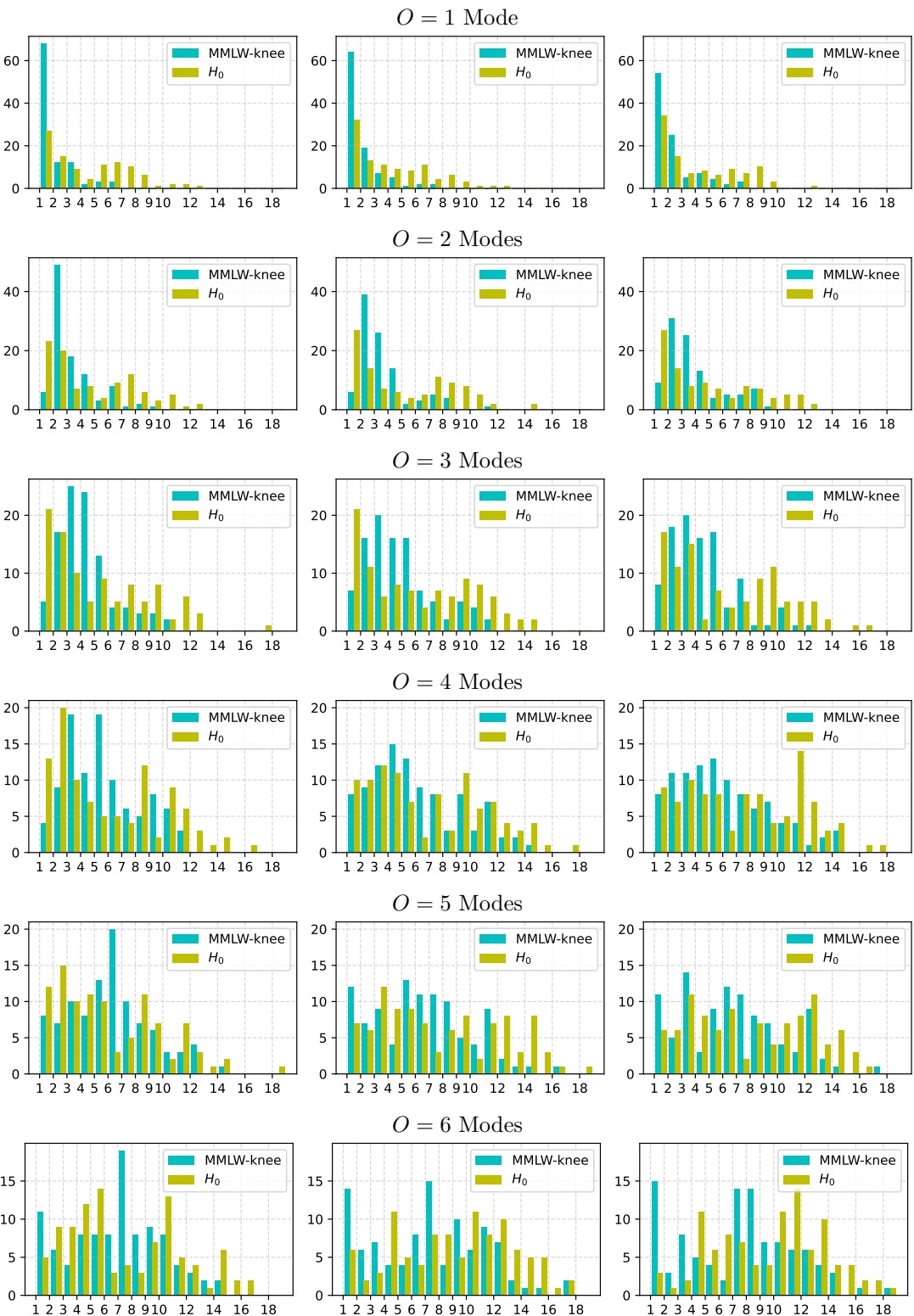

Figure 6: Number of landmarks selected by MMLW-ME method when $\mu = \mathcal{N}(0,1)$ and $\nu = 0.9\mu + 0.1\xi$ (or under $H_0$ where two samples are from $\frac{1}{2}\mu + \frac{1}{2}\nu$), where the outlier distribution $\xi = \frac{1}{O}\sum_{i=1}^{O}\mathcal{N}(a_i, \sigma^2)$ has $O \in \{1, 2, 3, 4, 5, 6\}$ modes at locations in the vector $\mathbf{a} = [-4, 4, -8, 8, -12, 12]$ and variance $\sigma^2 \in \{0.15, 1, 1.15\}$. The landmarks selected under $H_0$ created by a permutation are also shown. Samples sizes are $m = n = 250$.

### A.5.2 Additional Results for Experiment 1

Results comparing MLW (split and unsplit) and MMD to baselines for detecting upsamping of digit 1 are shown in Table 8. Type-1 error for $H_0$ cases are shown in Table 9. Table 10 shows results across different kernels choices of kernel and choices of $p$ for MLW.

Table 8: Test power (average and standard deviation) when upsampling of digit 1 in MNIST. MMD and MLW were run from scratch other values based on those reported by Wang et al. (2022). For a sample size, **best is bolded**, second best is underlined, and *the third best is italicized.*

| $m = n$ | MMD-NTK | MMD-D | ME | PW | KPW | MMD | MLW-split | MLW |
|---|---|---|---|---|---|---|---|---|
| 200 | 0.639±0.029 | 0.696±0.006 | 0.298±0.031 | 0.302±0.033 | 0.663±0.015 | 0.944±0.093 | *0.775±0.232* | **0.987±0.033** |
| 250 | 0.763±0.010 | 0.781±0.002 | 0.472±0.017 | 0.369±0.030 | 0.785±0.014 | 0.993±0.013 | *0.911±0.049* | **0.998±0.004** |
| 300 | 0.813±0.016 | 0.869±0.002 | 0.630±0.025 | 0.524±0.023 | 0.928±0.001 | 0.997±0.006 | *0.934±0.033* | **0.999±0.003** |
| 400 | 0.881±0.013 | 0.956±0.003 | 0.779±0.020 | 0.591±0.044 | 0.978±0.000 | **1.000±0.000** | *0.987±0.015* | **1.000±0.000** |
| 500 | 0.950±0.002 | 0.988±0.000 | 0.927±0.006 | 0.782±0.040 | **1.000±0.000** | **1.000±0.000** | *0.996±0.007* | **1.000±0.000** |
| Average | 0.809 | 0.858 | 0.621 | 0.513 | 0.870 | 0.987 | *0.921* | **0.997** |

Table 9: Average Type I (false-positive) error with standard deviation for two sample tests for digit 1 upsampling in MNIST dataset. MMD and MLW were run from scratch other values based on those reported by Wang et al. (2022).

| m=n | MMD-NTK | MMD-D | ME | PW | KPW | MMD | MLW-split | MLW |
|---|---|---|---|---|---|---|---|---|
| 200 | 0.057±0.010 | 0.056±0.006 | 0.044±0.003 | 0.056±0.004 | 0.061±0.005 | 0.042±0.087 | 0.060±0.017 | 0.030±0.050 |
| 250 | 0.051±0.003 | 0.060±0.001 | 0.065±0.002 | 0.046±0.003 | 0.048±0.002 | 0.041±0.064 | 0.049±0.023 | 0.022±0.019 |
| 300 | 0.068±0.006 | 0.055±0.003 | 0.059±0.007 | 0.056±0.002 | 0.053±0.001 | 0.051±0.044 | 0.063±0.015 | 0.065±0.061 |
| 400 | 0.049±0.007 | 0.058±0.002 | 0.041±0.002 | 0.061±0.006 | 0.056±0.006 | 0.072±0.071 | 0.053±0.031 | 0.055±0.037 |
| 500 | 0.061±0.006 | 0.054±0.004 | 0.060±0.002 | 0.049±0.003 | 0.047±0.004 | 0.035±0.034 | 0.054±0.031 | 0.034±0.030 |
| Average | 0.057 | 0.056 | 0.053 | 0.054 | 0.053 | 0.048 | 0.056 | 0.044 |

Table 10: Average test power (mean and standard deviation across 10 trials) across upsampling one digit at a time in MNIST for MLW with different choices of $p$ and kernel and MMD with different choices of kernels. Best method-kernel pair and ties (at 2 decimal digits) are bolded. Best settings for MLW are underlined.

| | MLW | | | | | | | | |
|---|---|---|---|---|---|---|---|---|---|
| $n$ | $p = 1$, Gau. | $p = 2$, Gau. | $p = 1$, Lap. | $p = 2$, Lap. | $p = 1$, Tri. | $p = 2$, Tri. | MMD, Gau. | MMD, Lap. | MMD, Tri. |
| 200 | 0.700±0.11 | 0.699±0.08 | 0.625±0.12 | 0.486±0.09 | 0.681±0.11 | 0.648±0.09 | 0.746±0.09 | **0.765±0.10** | 0.750±0.10 |
| 250 | 0.825±0.08 | 0.827±0.06 | 0.738±0.09 | 0.598±0.09 | 0.802±0.09 | 0.782±0.07 | 0.880±0.05 | **0.888±0.05** | 0.882±0.05 |
| 300 | 0.843±0.06 | 0.855±0.05 | 0.775±0.07 | 0.654±0.08 | 0.828±0.06 | 0.819±0.06 | 0.906±0.05 | **0.916±0.05** | 0.905±0.05 |
| 400 | 0.920±0.03 | 0.932±0.03 | 0.873±0.05 | 0.802±0.04 | 0.910±0.04 | 0.922±0.03 | **0.955±0.02** | 0.958±0.02 | 0.955±0.02 |
| 500 | 0.980±0.02 | 0.987±0.01 | 0.958±0.02 | 0.924±0.04 | 0.976±0.02 | 0.983±0.01 | **0.995±0.00** | 0.997±0.00 | 0.995±0.01 |

### A.5.3 High-Dimensional Gaussians

Using the same subset of methods from Experiment 1 we report results for synthetic data from Wang et al. (2022) to evaluate the ability of the tests to detect the shift between high-dimensional zero-mean Gaussian distributions with different covariances in Figure 7 for $m = n = 100$. We note that ME1 without splitting achieves the best performance and MLW achieves the second best for the first case of distinguishing data from isotropic Gaussian from a non-isotropic Gaussian with diagonal covariance. For the third case, where the second Gaussian has a general covariance, MLW and ME1 also perform well approaching the performance of projected Wasserstein (PW) approach. The split version of MLW performs similar to ME and ME1 on the first case and outperforms them on the second case.

### A.5.4 Additional Results for Experiment 2

Table 11 shows statistical power for downsampling of different classes in MNIST. Likewise, Table 12 shows

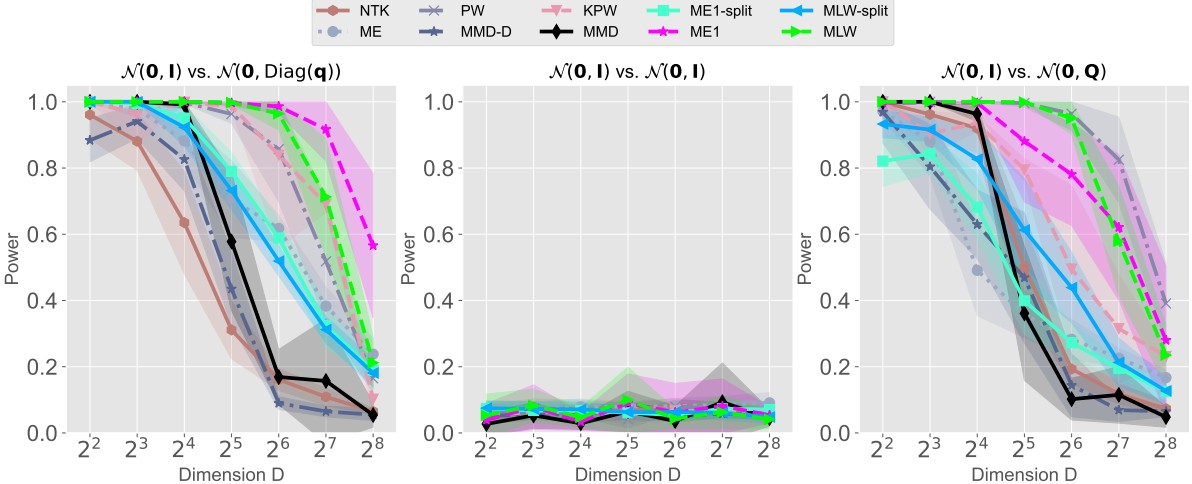

Figure 7: Power curves and Type I error at $\alpha = 0.05$ for baselines (Wang et al., 2022), ME1 (split and unsplit), and MLW (split and unsplit) when testing divergence of isotropic Gaussians with different variance in increasing dimension ($m = n = 100$). Tests use $B = 100$ permutations. In each trial, power and error are computed on 100 Monte Carlo generations. Plots are mean across 10 trials with filled area being one standard deviation from the mean.

statistical power for downsampling of different classes in CIFAR-10.

Figure 8 shows the relationship between statistical power and landmark accuracy for MLW for MNIST and CIFAR-10 datasets and a comparison of MMD to MLW.

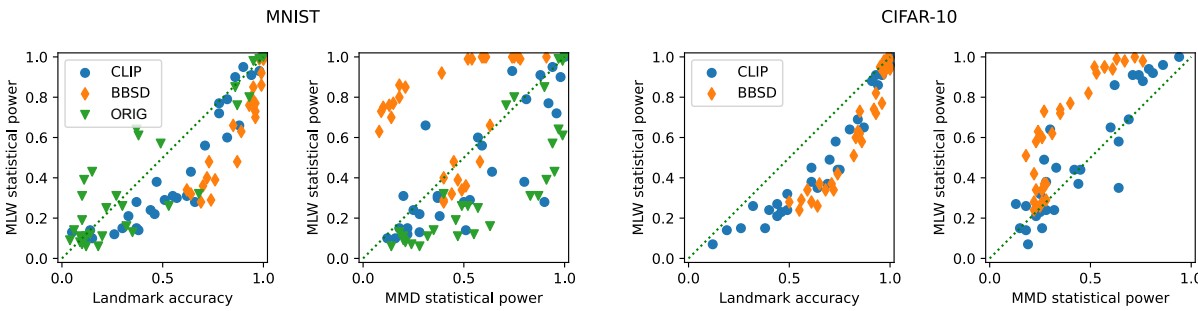

Figure 8: Relationship between performance values (statistical power at $\alpha = 0.05$ vs. landmark accuracy for minority class; MLW vs. MMD statistical power) on MNIST and CIFAR-10 from the tables.

Figure 9 shows landmarks selected by MLW and ME1 (and those with highest witness function magnitude for MMD) for cases of downsampling of different classes across different samples from test sets on CIFAR-10. Likewise, Figure 10 shows the results for MNIST.

The quantiative performance per class of the landmark accuracy when downsampling is in Table 13 and Table 14.

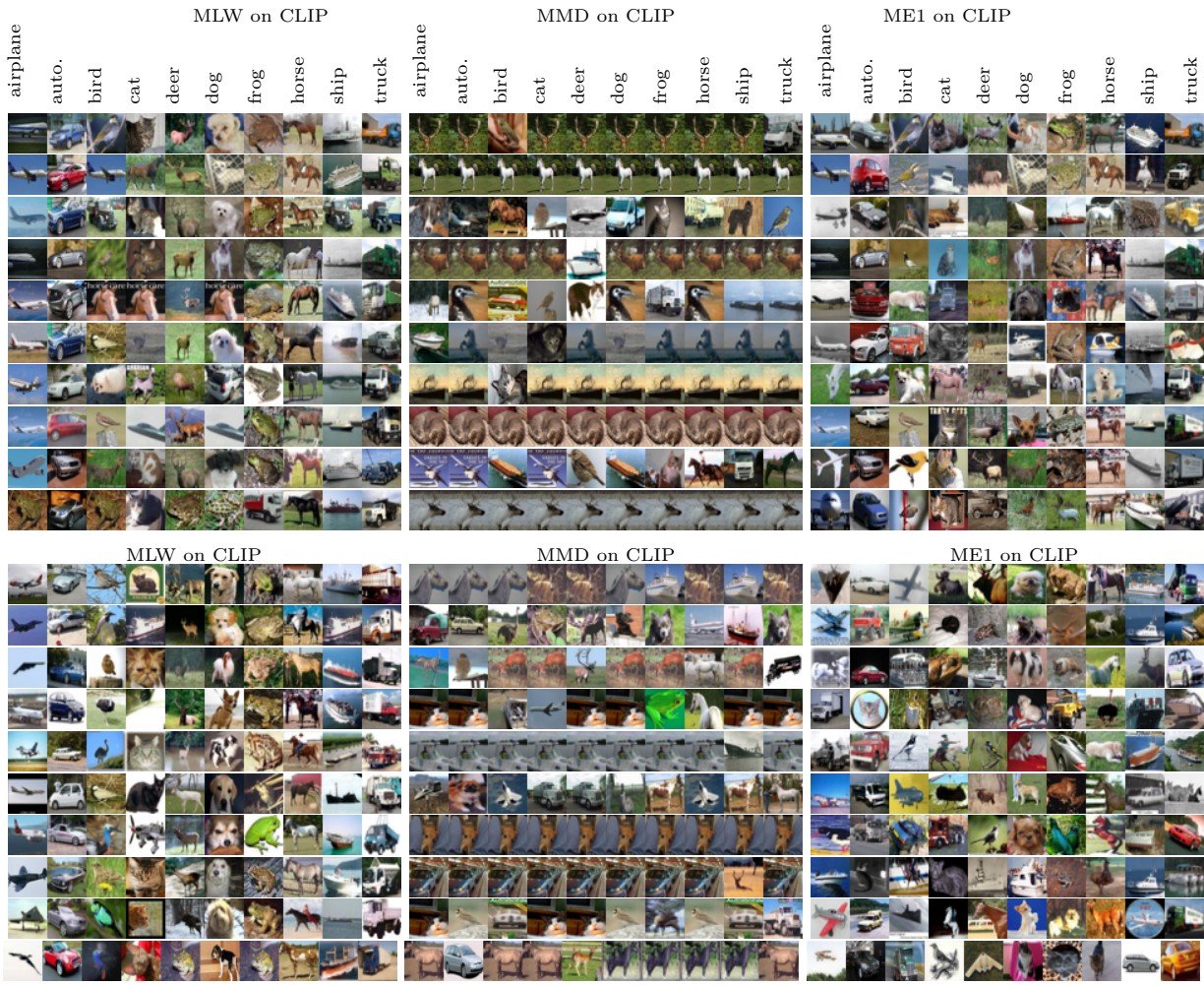

Figure 9: Landmarks chosen for CIFAR-10 when one class (indicated by column header) is a minority with prevalence 0.05 in the test set while the training set has uniform class distribution 0.1 prevalence. Sample size is $m = n = 2000$. Each row contains different trials, and each column uses the same training set, only the test set is modified. MLW with BBSD consistently selects landmarks from the minority class. Whereas MMD selects landmarks that may distinguish the training and test set, but are not sensitive to the imbalance. ME often selects reasonable landmarks, but not as frequently as MLW.

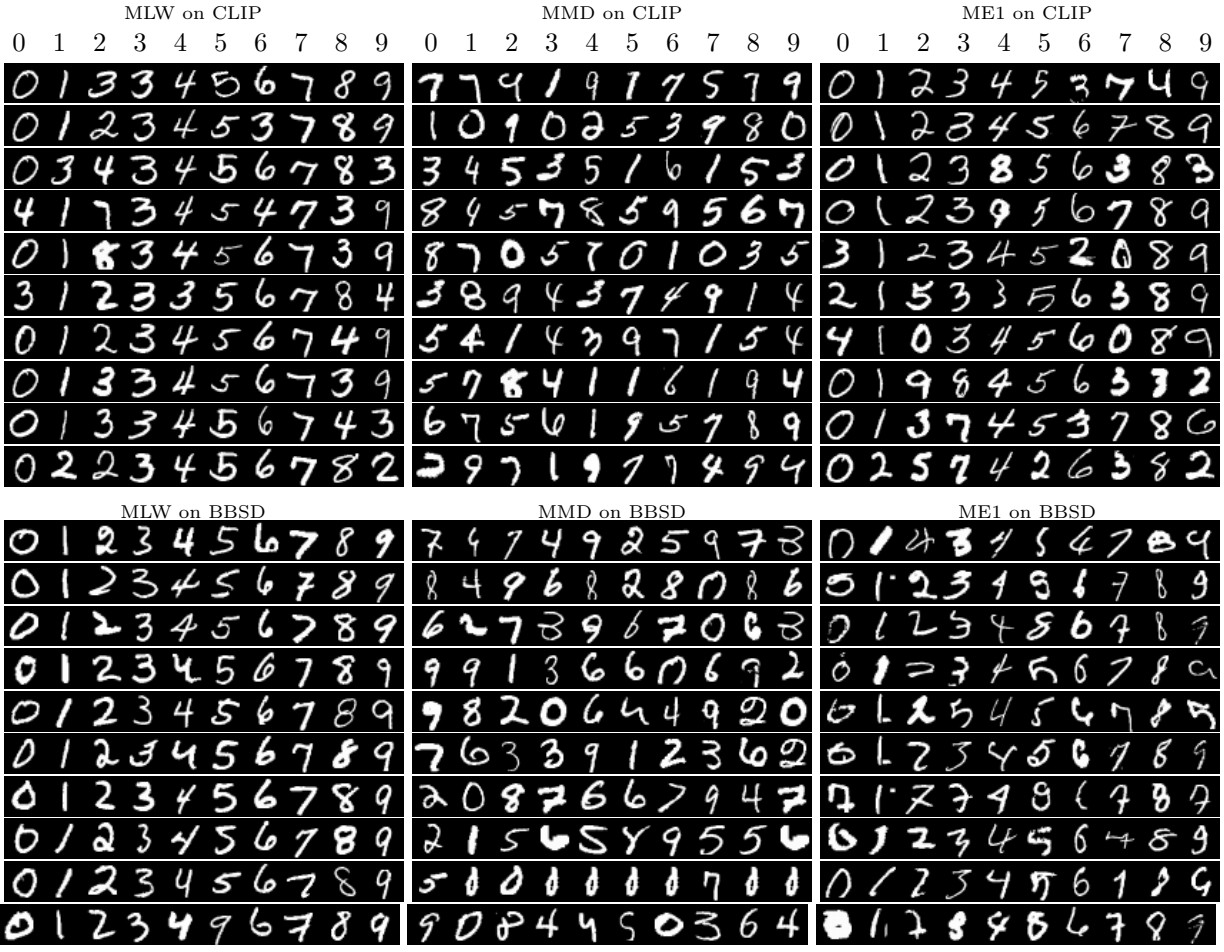

Figure 10: Landmarks chosen for MNIST when one class (indicated by column header) is a minority with prevalence 0.05 in the test set while the training set has uniform class distribution 0.1 prevalence. Sample size is $m = n = 2000$. Each row contains different trials, and each column uses the same training set, only the test set is modified.

Table 11: Statistical power at $\alpha = 0.05$ for detecting under-representation of one class in MNIST. After knockout the minority class has a prevalence of 5% compared to the other classes having 10.05%. Best method per class and average per dataset are bolded.

| $m = n$ | Repr. | Method | Class | | | | | | | | | | Average |
|---|---|---|---|---|---|---|---|---|---|---|---|---|---|
| | | | 0 | 1 | 2 | 3 | 4 | 5 | 6 | 7 | 8 | 9 | |
| 500 | Orig | MLW | 0.32 | **0.80** | 0.07 | 0.11 | 0.06 | 0.08 | 0.13 | 0.11 | 0.09 | 0.06 | 0.183 |
| | | MMD | **0.40** | 0.75 | **0.24** | **0.20** | **0.28** | **0.21** | 0.18 | **0.32** | **0.20** | 0.14 | **0.292** |
| | | ME1 | 0.39 | 0.49 | 0.18 | 0.18 | 0.24 | **0.21** | **0.19** | 0.26 | 0.12 | **0.18** | 0.244 |
| | CLIP | MLW | 0.14 | 0.22 | 0.10 | **0.66** | **0.31** | 0.15 | 0.24 | 0.15 | 0.13 | 0.10 | 0.220 |
| | | MMD | 0.21 | **0.28** | **0.12** | 0.31 | 0.20 | **0.22** | **0.25** | **0.18** | **0.28** | **0.16** | **0.221** |
| | | ME1 | **0.25** | 0.24 | 0.10 | 0.20 | 0.18 | 0.20 | 0.20 | 0.15 | 0.18 | 0.10 | 0.180 |
| | BBSD | MLW | 0.36 | 0.66 | 0.32 | 0.48 | 0.40 | 0.28 | 0.39 | 0.48 | 0.34 | 0.29 | 0.400 |
| | | MMD | 0.48 | 0.63 | 0.44 | 0.45 | 0.40 | 0.40 | 0.47 | 0.58 | 0.49 | 0.40 | 0.474 |
| | | ME1 | **0.52** | **0.70** | **0.46** | **0.62** | **0.53** | **0.46** | **0.64** | **0.61** | **0.51** | **0.46** | **0.551** |
| 1000 | Orig | MLW | **0.76** | **0.99** | 0.12 | 0.26 | 0.16 | 0.11 | 0.26 | 0.25 | 0.14 | 0.19 | 0.324 |
| | | MMD | 0.69 | 0.97 | **0.55** | **0.49** | **0.63** | **0.47** | **0.52** | **0.57** | **0.35** | **0.46** | **0.570** |
| | | ME1 | 0.69 | 0.87 | 0.42 | 0.45 | 0.61 | 0.37 | 0.37 | 0.47 | 0.23 | 0.34 | 0.482 |
| | CLIP | MLW | 0.28 | 0.43 | 0.12 | **0.93** | **0.60** | 0.30 | 0.56 | 0.31 | 0.14 | 0.21 | 0.388 |
| | | MMD | **0.55** | **0.64** | **0.22** | 0.74 | 0.57 | **0.37** | **0.59** | **0.37** | **0.51** | **0.38** | **0.494** |
| | | ME1 | 0.39 | 0.49 | 0.17 | 0.64 | 0.42 | 0.30 | 0.46 | 0.36 | 0.34 | 0.20 | 0.377 |
| | BBSD | MLW | 0.77 | 0.92 | 0.70 | 0.85 | 0.73 | 0.63 | 0.80 | 0.86 | 0.76 | 0.75 | 0.777 |
| | | MMD | 0.14 | 0.39 | 0.14 | 0.21 | 0.09 | 0.08 | 0.18 | 0.18 | 0.13 | 0.10 | 0.164 |
| | | ME1 | **0.90** | **0.96** | **0.84** | **0.92** | **0.87** | **0.91** | **0.87** | **0.95** | **0.85** | **0.87** | **0.894** |
| 2000 | Orig | MLW | 0.98 | **1.00** | 0.43 | 0.57 | 0.61 | 0.39 | 0.85 | 0.64 | 0.31 | 0.31 | 0.609 |
| | | MMD | **0.99** | **1.00** | **0.95** | **0.94** | **0.99** | **0.91** | **0.90** | **0.97** | **0.83** | **0.87** | **0.935** |
| | | ME1 | 0.98 | **1.00** | 0.90 | 0.88 | 0.96 | 0.88 | 0.81 | 0.92 | 0.59 | 0.65 | 0.857 |
| | CLIP | MLW | 0.72 | 0.90 | 0.29 | **1.00** | **0.91** | 0.79 | **0.95** | 0.77 | 0.28 | 0.38 | 0.699 |
| | | MMD | **0.98** | **0.98** | **0.53** | **1.00** | 0.88 | **0.81** | 0.94 | **0.92** | **0.90** | **0.80** | **0.874** |
| | | ME1 | 0.91 | 0.84 | 0.32 | 0.96 | 0.80 | 0.67 | 0.90 | 0.83 | 0.73 | 0.51 | 0.747 |
| | BBSD | MLW | **1.00** | **1.00** | **1.00** | 0.99 | **1.00** | **0.99** | **1.00** | **1.00** | **1.00** | 0.99 | 0.997 |
| | | MMD | 0.74 | 0.91 | 0.60 | 0.78 | 0.59 | 0.52 | 0.75 | 0.77 | 0.61 | 0.54 | 0.681 |
| | | ME1 | **1.00** | **1.00** | **1.00** | **1.00** | **1.00** | **0.99** | **1.00** | **1.00** | **1.00** | **1.00** | **0.999** |

Table 12: Statistical power at $\alpha = 0.05$ for detecting under-representation of one class in CIFAR-10. After knockout the minority class has a prevalence of 5% compared to the other classes having 10.05%. Best method per class and average per dataset are bolded.

| $m = n$ | Repr. | Method | airpl. | auto. | bird | cat | deer | dog | frog | horse | ship | truck | Average |
|---|---|---|---|---|---|---|---|---|---|---|---|---|---|
| | | | | | | | | Class | | | | | |
| 500 | Orig | MLW | **0.09** | 0.07 | 0.02 | 0.07 | **0.07** | 0.03 | **0.12** | 0.03 | 0.04 | 0.08 | 0.062 |
| | | MMD | 0.01 | 0.04 | **0.03** | 0.07 | 0.04 | 0.02 | 0.09 | 0.04 | 0.03 | 0.06 | 0.043 |
| | | ME1 | 0.07 | **0.09** | **0.03** | **0.09** | 0.04 | **0.09** | 0.11 | **0.05** | **0.07** | **0.09** | **0.073** |
| | CLIP | MLW | **0.25** | 0.24 | 0.07 | 0.14 | **0.26** | **0.15** | **0.27** | **0.49** | 0.21 | **0.38** | **0.246** |
| | | MMD | **0.25** | **0.28** | **0.19** | **0.18** | 0.18 | **0.15** | 0.13 | 0.27 | 0.23 | 0.28 | 0.214 |
| | | ME1 | 0.24 | 0.23 | 0.12 | 0.14 | 0.21 | 0.10 | 0.17 | 0.36 | **0.27** | 0.32 | 0.216 |
| | BBSD | MLW | 0.34 | 0.37 | 0.29 | **0.28** | 0.28 | 0.24 | **0.26** | 0.34 | 0.37 | **0.34** | 0.311 |
| | | MMD | 0.26 | 0.28 | 0.28 | 0.26 | 0.22 | 0.22 | 0.25 | 0.26 | 0.27 | 0.23 | 0.253 |
| | | ME1 | **0.37** | **0.44** | **0.31** | **0.28** | **0.35** | **0.31** | 0.25 | **0.36** | **0.43** | 0.33 | **0.343** |
| 1000 | Orig | MLW | **0.11** | **0.10** | **0.09** | **0.10** | **0.11** | 0.05 | **0.14** | 0.04 | **0.08** | 0.04 | **0.086** |
| | | MMD | 0.05 | 0.05 | 0.08 | 0.08 | 0.07 | 0.03 | **0.14** | 0.05 | 0.03 | 0.05 | 0.063 |
| | | ME1 | 0.05 | **0.10** | 0.06 | **0.10** | 0.08 | **0.10** | 0.09 | **0.08** | 0.06 | **0.05** | 0.077 |
| | CLIP | MLW | **0.47** | 0.44 | 0.15 | **0.32** | **0.45** | 0.24 | **0.64** | **0.86** | 0.37 | **0.65** | **0.459** |
| | | MMD | 0.42 | **0.45** | **0.26** | 0.25 | 0.33 | **0.32** | 0.30 | 0.62 | **0.44** | 0.60 | 0.399 |
| | | ME1 | 0.39 | **0.45** | 0.20 | 0.24 | 0.37 | 0.19 | 0.27 | 0.65 | 0.42 | 0.61 | 0.379 |
| | BBSD | MLW | 0.74 | **0.77** | 0.60 | 0.42 | 0.58 | 0.51 | 0.62 | 0.73 | **0.72** | **0.63** | 0.632 |
| | | MMD | 0.35 | 0.40 | 0.26 | 0.22 | 0.23 | 0.18 | 0.31 | 0.27 | 0.35 | 0.24 | 0.281 |
| | | ME1 | **0.80** | **0.77** | **0.73** | **0.61** | **0.64** | **0.63** | **0.67** | **0.75** | **0.72** | 0.59 | **0.691** |
| 2000 | Orig | MLW | **0.14** | 0.13 | 0.09 | 0.18 | 0.22 | 0.10 | 0.31 | 0.11 | **0.11** | 0.10 | 0.149 |
| | | MMD | 0.05 | 0.13 | **0.12** | **0.23** | **0.29** | 0.12 | **0.33** | **0.14** | **0.11** | 0.09 | **0.161** |
| | | ME1 | 0.09 | **0.21** | 0.07 | 0.19 | 0.14 | **0.13** | 0.22 | 0.11 | **0.11** | **0.20** | 0.147 |
| | CLIP | MLW | **0.87** | **0.91** | 0.35 | **0.69** | **0.91** | 0.58 | **0.94** | **1.00** | **0.92** | **0.96** | **0.813** |
| | | MMD | 0.76 | 0.71 | **0.64** | **0.69** | 0.74 | **0.64** | 0.79 | 0.94 | 0.81 | 0.86 | 0.758 |
| | | ME1 | 0.81 | 0.80 | 0.46 | 0.55 | 0.71 | 0.49 | 0.75 | 0.96 | 0.82 | 0.89 | 0.724 |
| | BBSD | MLW | **0.98** | **1.00** | 0.95 | 0.82 | 0.92 | 0.95 | 0.94 | **0.99** | **0.98** | **0.95** | 0.948 |
| | | MMD | 0.76 | 0.72 | 0.57 | 0.49 | 0.53 | 0.57 | 0.61 | 0.63 | 0.67 | 0.52 | 0.607 |
| | | ME1 | **0.98** | **1.00** | **0.96** | **0.89** | **0.95** | **0.98** | **0.95** | **0.99** | **0.98** | 0.91 | **0.959** |

Table 13: Accuracy of the landmark corresponding to knockout distribution shift on CIFAR-10. After knockout the minority class has a prevalence of 5% compared to the other classes having 10.05%. Accuracy is calculated across 100 random draws.

| $m = n$ | Repr. | Method | airplane | auto. | bird | cat | deer | dog | frog | horse | ship | truck | Average |
|---|---|---|---|---|---|---|---|---|---|---|---|---|---|---|
| | | | | | | | | Class | | | | | | |
| 500 | Orig | MLW | **0.28** | 0.05 | 0.09 | 0.06 | 0.06 | 0.02 | 0.07 | 0.07 | **0.06** | 0.04 | 0.080 |
| | | MMD | **0.28** | 0.03 | **0.19** | **0.08** | **0.10** | 0.05 | **0.10** | 0.08 | 0.05 | 0.02 | **0.098** |
| | | ME1 | 0.06 | **0.09** | 0.02 | **0.08** | 0.07 | **0.14** | 0.02 | **0.10** | 0.03 | **0.08** | 0.069 |
| | CLIP | MLW | 0.46 | 0.49 | 0.12 | 0.19 | 0.32 | 0.26 | **0.44** | **0.70** | 0.44 | **0.61** | 0.403 |
| | | MMD | 0.15 | 0.12 | 0.10 | 0.10 | 0.11 | 0.13 | 0.08 | 0.11 | 0.06 | 0.14 | 0.110 |
| | | ME1 | **0.49** | **0.53** | **0.38** | **0.37** | **0.44** | **0.32** | 0.36 | 0.67 | **0.59** | 0.49 | **0.464** |
| | BBSD | MLW | **0.68** | **0.71** | **0.56** | **0.50** | **0.61** | **0.55** | **0.64** | **0.72** | **0.65** | **0.59** | **0.621** |
| | | MMD | 0.13 | 0.21 | 0.17 | 0.17 | 0.18 | 0.09 | 0.27 | 0.13 | 0.23 | 0.12 | 0.170 |
| | | ME1 | 0.34 | 0.34 | 0.28 | 0.24 | 0.23 | 0.35 | 0.22 | 0.28 | 0.31 | 0.20 | 0.279 |
| 1000 | Orig | MLW | 0.25 | 0.03 | 0.10 | **0.09** | **0.15** | 0.05 | 0.06 | **0.07** | **0.13** | 0.02 | 0.095 |
| | | MMD | **0.30** | 0.03 | **0.23** | 0.04 | 0.13 | 0.04 | **0.11** | 0.05 | 0.04 | 0.02 | **0.099** |
| | | ME1 | 0.14 | **0.05** | 0.07 | 0.07 | 0.07 | **0.16** | 0.05 | 0.05 | **0.13** | **0.13** | 0.092 |
| | CLIP | MLW | **0.74** | 0.75 | 0.38 | 0.49 | 0.61 | 0.40 | **0.80** | **0.94** | 0.69 | **0.87** | **0.667** |
| | | MMD | 0.17 | 0.14 | 0.12 | 0.14 | 0.10 | 0.13 | 0.05 | 0.15 | 0.10 | 0.13 | 0.123 |
| | | ME1 | 0.67 | **0.77** | **0.62** | **0.55** | **0.68** | **0.42** | 0.62 | 0.86 | **0.73** | 0.67 | 0.659 |
| | BBSD | MLW | **0.92** | **0.96** | **0.83** | **0.74** | **0.86** | **0.82** | **0.85** | **0.85** | **0.93** | **0.84** | **0.860** |
| | | MMD | 0.16 | 0.18 | 0.20 | 0.15 | 0.18 | 0.10 | 0.21 | 0.21 | 0.22 | 0.22 | 0.183 |
| | | ME1 | 0.43 | 0.56 | 0.28 | 0.31 | 0.33 | 0.37 | 0.37 | 0.33 | 0.42 | 0.30 | 0.370 |
| 2000 | Orig | MLW | **0.36** | **0.05** | 0.10 | 0.03 | **0.23** | **0.06** | **0.13** | 0.01 | **0.13** | 0.02 | **0.112** |
| | | MMD | 0.34 | 0.01 | **0.25** | 0.04 | 0.15 | 0.04 | 0.12 | 0.03 | 0.04 | 0.01 | 0.103 |
| | | ME1 | 0.15 | 0.04 | 0.06 | **0.08** | 0.09 | **0.06** | 0.02 | **0.05** | 0.08 | **0.08** | 0.071 |
| | CLIP | MLW | **0.91** | 0.96 | 0.64 | 0.84 | **0.93** | 0.73 | **0.98** | **1.00** | 0.96 | **1.00** | **0.895** |
| | | MMD | 0.17 | 0.16 | 0.10 | 0.13 | 0.11 | 0.10 | 0.05 | 0.08 | 0.11 | 0.10 | 0.111 |
| | | ME1 | 0.86 | **0.97** | **0.79** | **0.87** | 0.91 | **0.74** | 0.83 | 0.96 | 0.93 | 0.86 | 0.872 |
| | BBSD | MLW | **0.97** | **0.99** | **0.99** | **0.93** | **0.97** | **0.95** | **1.00** | **1.00** | **0.99** | **1.00** | **0.979** |
| | | MMD | 0.19 | 0.23 | 0.21 | 0.14 | 0.16 | 0.14 | 0.16 | 0.25 | 0.20 | 0.22 | 0.190 |
| | | ME1 | 0.44 | 0.64 | 0.27 | 0.30 | 0.35 | 0.45 | 0.36 | 0.44 | 0.49 | 0.32 | 0.406 |

Table 14: Accuracy of the landmark corresponding to knockout distribution shift on MNIST. After knockout the minority class has a prevalence of 5% compared to the other classes having 10.05%. Accuracy is calculated across 100 random draws.

| $m=n$ | Repr. | Method | 0 | 1 | 2 | 3 | 4 | 5 | 6 | 7 | 8 | 9 | Average |
|---|---|---|---|---|---|---|---|---|---|---|---|---|---|
| | | | | | | Class | | | | | | | |
| 500 | Orig | MLW | **0.68** | **0.93** | 0.10 | 0.20 | **0.18** | 0.09 | **0.35** | **0.12** | 0.04 | **0.12** | **0.281** |
| | | MMD | 0.45 | 0.19 | 0.02 | 0.06 | 0.05 | 0.05 | 0.07 | 0.04 | 0.08 | **0.12** | 0.113 |
| | | ME1 | 0.22 | 0.18 | **0.15** | **0.22** | 0.13 | **0.10** | 0.14 | 0.06 | **0.13** | 0.09 | 0.142 |
| | CLIP | MLW | **0.38** | **0.46** | 0.13 | **0.88** | **0.55** | 0.30 | **0.44** | **0.37** | 0.05 | 0.15 | **0.371** |
| | | MMD | 0.12 | 0.07 | 0.06 | 0.10 | 0.08 | 0.19 | 0.07 | 0.23 | 0.12 | 0.13 | 0.117 |
| | | ME1 | **0.38** | 0.43 | **0.22** | 0.36 | 0.39 | **0.33** | 0.30 | 0.23 | **0.17** | **0.24** | 0.305 |
| | BBSD | MLW | 0.70 | 0.85 | 0.64 | **0.87** | 0.72 | 0.69 | 0.76 | 0.73 | 0.62 | **0.74** | 0.732 |
| | | MMD | 0.11 | 0.08 | 0.16 | 0.14 | 0.12 | 0.15 | 0.06 | 0.18 | 0.18 | 0.12 | 0.130 |
| | | ME1 | **0.72** | **0.89** | **0.70** | 0.84 | **0.80** | **0.75** | **0.79** | **0.80** | **0.69** | **0.74** | **0.772** |
| 1000 | Orig | MLW | **0.87** | **0.98** | 0.14 | **0.30** | **0.32** | **0.09** | **0.53** | **0.22** | 0.06 | 0.10 | **0.361** |
| | | MMD | 0.47 | 0.20 | 0.01 | 0.07 | 0.06 | 0.03 | 0.04 | 0.03 | 0.06 | **0.12** | 0.109 |
| | | ME1 | 0.23 | 0.24 | **0.21** | 0.29 | 0.26 | 0.07 | 0.08 | 0.05 | **0.12** | 0.06 | 0.161 |
| | CLIP | MLW | **0.66** | 0.64 | 0.26 | **0.98** | **0.82** | **0.57** | **0.71** | **0.62** | 0.14 | **0.33** | **0.573** |
| | | MMD | 0.02 | 0.06 | 0.07 | 0.08 | 0.04 | 0.18 | 0.05 | 0.21 | 0.09 | 0.11 | 0.091 |
| | | ME1 | 0.48 | **0.68** | **0.33** | 0.40 | 0.44 | 0.50 | 0.45 | 0.29 | **0.36** | **0.33** | 0.426 |
| | BBSD | MLW | **0.96** | **0.99** | **0.96** | **0.95** | **0.96** | **0.89** | **0.94** | **0.99** | **0.93** | **0.95** | **0.952** |
| | | MMD | 0.09 | 0.08 | 0.11 | 0.11 | 0.10 | 0.07 | 0.10 | 0.13 | 0.15 | 0.09 | 0.103 |
| | | ME1 | 0.84 | 0.96 | 0.84 | 0.88 | **0.96** | 0.87 | 0.93 | 0.97 | 0.85 | 0.81 | 0.891 |
| 2000 | Orig | MLW | **0.95** | **1.00** | **0.15** | **0.49** | 0.38 | **0.11** | **0.86** | **0.37** | 0.10 | **0.27** | **0.468** |
| | | MMD | 0.48 | 0.25 | 0.01 | 0.02 | 0.07 | 0.00 | 0.04 | 0.04 | 0.07 | 0.09 | 0.107 |
| | | ME1 | 0.21 | 0.33 | 0.11 | 0.17 | **0.40** | 0.03 | 0.10 | 0.08 | **0.23** | 0.08 | 0.174 |
| | CLIP | MLW | **0.78** | **0.86** | **0.51** | **0.99** | **0.94** | **0.82** | **0.90** | **0.78** | 0.37 | 0.47 | **0.742** |
| | | MMD | 0.02 | 0.09 | 0.07 | 0.13 | 0.04 | 0.24 | 0.10 | 0.15 | 0.16 | 0.13 | 0.113 |
| | | ME1 | 0.62 | 0.85 | 0.49 | 0.49 | 0.47 | 0.54 | 0.56 | 0.27 | **0.51** | **0.58** | 0.538 |
| | BBSD | MLW | **1.00** | **1.00** | **1.00** | **1.00** | **1.00** | **0.98** | **1.00** | **1.00** | **1.00** | **1.00** | **0.998** |
| | | MMD | 0.16 | 0.09 | 0.09 | 0.11 | 0.13 | 0.07 | 0.10 | 0.13 | 0.15 | 0.08 | 0.111 |
| | | ME1 | 0.76 | 0.98 | 0.84 | 0.83 | 0.95 | 0.88 | 0.94 | 0.92 | 0.88 | 0.82 | 0.880 |

In the 'Failing Loudly' framework by Rabanser et al. (2019), applying MMD on BBSD initially showed better power; however, it didn't perform well for other representations. We noticed that the Gaussian kernel definition is not consistent with the standard definition. Specifically, the implementation uses the kernel $\kappa(x, y) = \exp(-\alpha\|x - y\|^2)$, where $\alpha = \text{median}(\{\|x_i - y_j\|\}_{i=1, j=1}^{m,n})$ is the median distance across pairs between the samples. This mistake means the kernel size $\sigma = \frac{1}{\sqrt{2\alpha}}$ is inversely related to the median distance and is not homogeneous with respect to the input such that the scale of the input matters, which explains its inconsistent performance across learning representations as shown in Table 15.

Table 15: Statistical power for detecting imbalanced distributions on CIFAR-10 using various learning representations (LR). After knockout the minority class has a prevalence of 5% compared to the other classes having 10.05%. Power is calculated across 100 random draws, hypothesis test use $\alpha = 0.05$ significance level and perform 100 random shuffling to generate the surrogate null distribution. Values are average power across all 10 classes. The best performance per learning representation and sample size is bolded. 'Failing' refers to implementation (Rabanser et al., 2019) where MMD uses a Gaussian kernel with an incorrect usage of the median distance.

| LR | Methods | MNIST Sample size $m = n$ | | | CIFAR-10 Sample size $m = n$ | | |
|---|---|---|---|---|---|---|---|
| | | 500 | 1000 | 2000 | 500 | 1000 | 2000 |
| Orig | MLW | 0.18 | 0.32 | 0.61 | **0.06** | **0.09** | 0.15 |
| | MMD | **0.29** | **0.57** | **0.94** | 0.04 | 0.06 | **0.16** |
| | Failing | 0.17 | 0.23 | 0.36 | 0.05 | 0.06 | 0.03 |
| CLIP | MLW | 0.22 | 0.39 | 0.70 | 0.24 | **0.46** | **0.81** |
| | MMD | **0.22** | **0.49** | **0.87** | 0.21 | 0.40 | 0.76 |
| | Failing | 0.01 | 0.00 | 0.00 | **1.00** | 0.00 | 0.00 |
| BBSD | MLW | 0.40 | **0.78** | **1.00** | **0.31** | **0.63** | **0.95** |
| | MMD | **0.48** | 0.17 | 0.68 | 0.25 | 0.28 | 0.61 |
| | Failing | 0.28 | 0.56 | 0.92 | 0.25 | 0.44 | 0.86 |

### A.5.5 Details and Additional Results for Higgs (Experiment 3)

For the experiments, we use the dataset and two-sample testing implementation released along with the AutoML two-sample test approach (Kübler et al., 2022), which was adapted from Liu et al. (2020). Contrary to our notation of sample size, in this implementation the sample size refers to the sizes of the train and test sets separately, rather than to their union. We apply MLW with train-test split, $p = 2$ with a Gaussian kernel, using 300 permutations. We also apply MMLW with $q = 13 \approx 2000^{1/3}$ and the same settings. While the naive implementations of MLW and MMLW did not scale to a sample size of $n^{\text{Tr}} = n^{\text{Te}} = 10000$ due to limited memory, batch-based versions that do not keep the entire kernel matrix in memory can scale. Additionally, for speed the median heuristic can be approximated by taking a random subset of the pooled sample. We calculate power from 100 runs and compute average and standard error for 5 fixed seeds.

Baseline methods include ME (Jitkrittum et al., 2016), SCF (Chwialkowski et al., 2015), classifier-based two-sample tests C2ST-S (Lopez-Paz & Oquab, 2016), C2ST-L (Cheng & Cloninger, 2022), MMD-D (Liu et al., 2020), and MMD-D (Liu et al., 2020). Results for baselines are taken from Liu et al. (2020) and the 10 minute AutoML copied from Kübler et al. (2022). As shown in Table 16 MLW with $p = 2$ and Gaussian kernel generally outperforms the SCF method (except for $n = 1000$ and $n = 2000$) but performs worse than the classifier-based two-sample tests and MMD. This indicates that the Higgs cases are not localized in the 4-dimensional feature space chosen by Chwialkowski et al. (2015). In the original work (Chwialkowski et al., 2015), the smoothed characteristic function (SCF) method outperformed MMD, and had much higher power compared to what is reported in Liu et al. (2020). In any case, "when distinguishing signatures of the Higgs boson from background noise, we observe that a test based on differences in smoothed empirical characteristic functions outperforms the quadratic-time MMD". MMLW has superior performance to MLW and on many sample sizes outperforms the ME approach (Jitkrittum et al., 2016) that uses 10 continuously optimized inducing points. We also use MLW on top of the kernel optimized by MMD-D. This creates large increases in performance, which do not match MMD-D, but outperform ME and AutoML (up to $n = 6000$). As the

MMD-D implementation did not use batching for the kernel construction, out-of-memory was experience for sample sizes of 8000 and 10000.

Table 16: Power (average and standard error) at $\alpha = 0.05$ for the Higgs dataset. The column headings indicate $n^{\text{Tr}} = n^{\text{Te}}$. MLW with $\kappa_{\text{MMD-D}}$ corresponds to applying MLW on the kernel optimized by MMD-D. Best result in each column is bolded, second best is underlined.

| | 1000 | 2000 | 3000 | 4000 | 5000 | 6000 | 8000 | 10000 |
|---|---|---|---|---|---|---|---|---|
| MLW | 0.048±0.007 | 0.086±0.010 | 0.150±0.017 | 0.206±0.010 | 0.300±0.011 | 0.438±0.023 | 0.612±0.022 | 0.764±0.022 |
| w/$\kappa_{\text{MMD-D}}$ | 0.092±0.008 | 0.182±0.015 | 0.306±0.017 | 0.514±0.037 | 0.604±0.025 | 0.686±0.022 | - | - |
| MMLW | 0.068±0.012 | 0.146±0.011 | 0.202±0.014 | 0.302±0.019 | 0.376±0.011 | 0.522±0.017 | 0.726±0.012 | 0.876±0.014 |
| ME | 0.120±0.007 | 0.165±0.019 | 0.197±0.012 | - | 0.410±0.041 | - | 0.691±0.067 | 0.786±0.041 |
| w/MMLW | 0.106±0.011 | 0.234±0.025 | **0.422±0.017** | **0.604±0.028** | **0.718±0.020** | **0.856±0.017** | 0.950±0.004 | 0.986±0.006 |
| SCF | 0.095±0.022 | 0.130±0.026 | 0.142±0.025 | - | 0.261±0.044 | - | 0.467±0.038 | 0.603±0.066 |
| C2ST-S | 0.082±0.015 | 0.183±0.032 | 0.257±0.049 | - | 0.592±0.037 | - | 0.892±0.029 | 0.974±0.007 |
| C2ST-L | 0.097±0.014 | 0.232±0.017 | 0.399±0.058 | - | 0.447±0.045 | - | 0.878±0.020 | 0.985±0.005 |
| MMD-O | **0.132±0.005** | 0.291±0.012 | 0.376±0.022 | - | 0.659±0.018 | - | 0.923±0.013 | **1.000±0.000** |
| MMD-D | 0.113±0.013 | **0.304±0.035** | 0.403±0.050 | - | 0.699±0.047 | - | **0.952±0.024** | **1.000±0.000** |
| AutoML | 0.09±0.01 | 0.17±0.02 | 0.25±0.02 | 0.40±0.02 | 0.63±0.02 | 0.80±0.02 | 0.93±0.01 | 0.99±0.00 |

We ran MLW with different choices of $p \in \{1, 2\}$ and kernel, and find that $p = 2$ and triangular kernel is consistently the best for the Higgs dataset. The results are shown in Table 18.

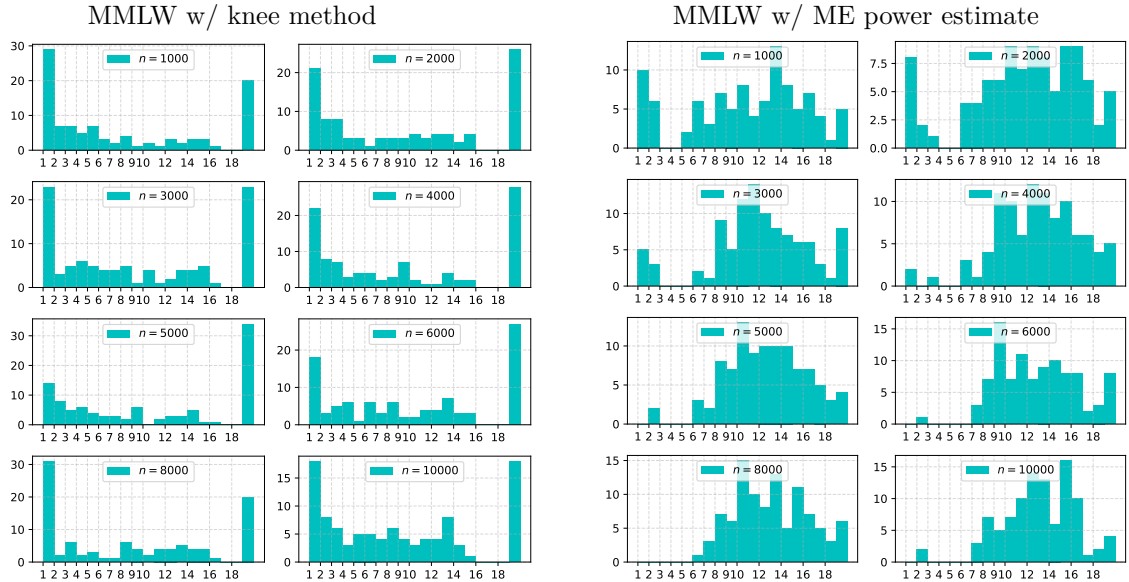

Figure 11: Number of landmarks selected by for the Higgs dataset. Here $n$ indicates $n^{\text{Tr}} = n^{\text{Te}}$. (Left) MMLW-knee using Kneedle (Satopaa et al., 2011) with default parameters under the concave increasing assumption. (Right) MMLW-ME.

Run times for MLW and MMLW using Python 3 Google Compute Engine (CPU) with 12.7 GB RAM on the Higgs dataset is in Table 17.

Table 17: Run times for the 4-dimensional Higgs dataset across different sample sizes. MMLW-Split uses $q = 13$ and MMLW-Knee uses $q = 20$ and then selects a smaller number of landmarks. MMLW-Knee and ME with MMLW uses optimized code with random sampling for the median heuristic when $n > 4000$ and batch-based computation of columns of the kernel matrix and the divergence. This optimized code is used for all methods for $n = 10000$. (Note that the sample size of train and test are equal to the number given, meaning each total sample size $n^{\mathrm{Tr}} + n^{\mathrm{Te}}$ is twice as large).

| Sample size $n^{\mathrm{Tr}} = n^{\mathrm{Te}}$: | 1000 | 2000 | 3000 | 4000 | 5000 | 6000 | 8000 | 10000 |
|---|---|---|---|---|---|---|---|---|
| MLW-Split run time (s): | 1.71 | 2.74 | 4.67 | 8.36 | 12.20 | 17.68 | 33.74 | 25.03 |
| MMLW-Split run time (s): | 1.15 | 2.68 | 6.27 | 9.28 | 15.51 | 21.09 | 41.06 | 21.08 |
| MMLW-Knee run time (s): | 3.16 | 5.65 | 8.55 | 11.66 | 15.22 | 19.49 | 29.35 | 42.27 |
| MMLW-Knee train-only run time (s): | 2.56 | 4.46 | 6.83 | 9.71 | 12.36 | 15.58 | 23.10 | 33.54 |
| ME w/MMLW run time (s): | 2.87 | 5.04 | 7.49 | 10.44 | 13.45 | 18.03 | 26.84 | 41.98 |

Table 18: Power (average and standard error) at $\alpha = 0.05$ for MLW for $p \in \{1, 2\}$ and different kernels (all using the median heuristic) on the Higgs dataset.

| $n^{\mathrm{Tr}} = n^{\mathrm{Te}}$ | $p = 1$, Gau. | $p = 2$, Gau. | $p = 1$, Lap. | $p = 2$, Lap. | $p = 1$, Tri. | $p = 2$, Tri. |
|---|---|---|---|---|---|---|
| 1000 | 0.058±0.009 | 0.048±0.007 | 0.048±0.007 | 0.056±0.011 | 0.064±0.012 | 0.060±0.008 |
| 2000 | 0.076±0.007 | 0.086±0.010 | 0.080±0.013 | 0.106±0.009 | 0.088±0.007 | 0.114±0.015 |
| 3000 | 0.110±0.009 | 0.150±0.017 | 0.102±0.015 | 0.170±0.012 | 0.118±0.009 | 0.170±0.014 |
| 4000 | 0.190±0.010 | 0.206±0.010 | 0.194±0.014 | 0.254±0.013 | 0.206±0.015 | 0.248±0.011 |
| 5000 | 0.262±0.008 | 0.300±0.011 | 0.278±0.019 | 0.310±0.013 | 0.282±0.013 | 0.344±0.007 |
| 6000 | 0.370±0.026 | 0.438±0.023 | 0.398±0.020 | 0.410±0.034 | 0.408±0.030 | 0.488±0.024 |
| 8000 | 0.546±0.013 | 0.612±0.022 | 0.572±0.008 | 0.568±0.027 | 0.552±0.022 | 0.682±0.032 |

