# OpenReview forum: "Finding Landmarks of Covariate Shift with the Max-Sliced Kernel Wasserstein Distance"
_TMLR — Rejected by TMLR_

### Review · Reviewer_Ejpb · 2026-02-13

**Summary Of Contributions:**

This paper proposes a metric for detecting covariate shift. The metric is called max landmark kernel Wasserstein distance (MLW). It is derived from the kernel max-sliced Wasserstein distance (KMS), which measures the Wasserstein distance between two distributions with respect the worse-case member of an RKHS. Specifically, MLW restricts this member to be $k_z$ for some $z$. The authors show how to compute the MLW distance on a set of training data. The authors also propose the multiple MLW distance that can use multiple $z$ in the MLW. The authors test their proposed metric on toy data, MNIST and CIFAR-10, by upsampling or downsampling some classes.

**Audience:**

No

**Audience Explanation:**

The proposed MLW distance is nothing more than a special case of the KMS distance proposed in a previous paper, and the authors do not compare between these two distances and elaborate on how weaker MLW is compared to KMS. While the method to compute MLW is somewhat interesting, given that the usefulness of the MLW itself is questionable, I don't see how the paper is interesting to the TMLR audience. I cannot recommend accepting this paper unless the authors can convince me that the proposed distance is useful in at least some realistic applications.

**Claims And Evidence:**

No

**Claims Explanation:**

The authors claim that the proposed MLW distance is better at detecting covariate shift. However, neither the theoretical evidence nor the empirical evidence is strong enough for this claim. On the theoretical side, the authors do not explain how much weaker MLW is compared to the original KMS distance that can use any member in the RKHS as the witness function. Moreover, although the KMS distance seems to be an interesting formulation, it is unclear to me how it is useful for detecting covariate shift. It largely depends on the kernel, yet the authors do not discuss how to choose the kernel. On the empirical side, for MNIST and CIFAR-10, the authors create distribution shift only by upsampling or downsampling some classes, which only represents one type of distribution shift. On the Higgs dataset where the distribution shift seems more realistic, the MLW is not competitive compared to other methods. Thus, I don't think the evidence is strong enough to support the main claim of this submission.

**Requested Changes:**

1. MLW is KMS with the witness function restricted to $k_z$. Compare between MLW and KMS. How much weaker is MLW? For example, if MLW is small but KMS is large, then what can you infer from this?
2. Add more background on KMS. Specifically, explain why KMS is useful for detecting covariate shift. What type of shift is it good at detecting? What type of shift cannot be detected by KMS?
3. How to select the kernel in KMS? Why does the Gaussian kernel make sense? What assumptions do you make when you select the Gaussian kernel?
4. In Multiple MLW (MMLW), if I choose a sufficient number of landmarks, will its power be as strong as the original KMS with no constraint on the witness function?
5. Add more experiments with more realistic settings of distribution shift.

---

> ### Author Response · Authors · 2026-03-24
>
> We thank Reviewer Ejpb for the time in reviewing our paper, the feedback has motivated some additional results that will strengthen our work. We plan to revise the paper, but haven't finalized the additional results, and note that the response window is ending.
>
> Regarding the statement, “it is unclear to me how it [KMS distance] is useful for detecting covariate shift.  It largely depends on the kernel, yet the authors do not discuss how to choose the kernel“, we note that KMS has been shown in previous (AISTATS 2022) and recent work (ICML 2025) to detect distribution shifts in challenging circumstances; however we have not been able to find an implementation of the newer algorithm (reduced rank semidefinite relaxation) for KMS. In theory, the KMS test statistic for the $p=2$ Wasserstein distance upper bounds MMD as it compares the full distribution of sliced values rather than the difference of means. Additionally, we note that the Gaussian kernel for kernel-based two-sample testing is theoretically motivated by its bounded and characteristic nature and has been used in the majority of prior work. (Some more recent work aggregates or fuses multiple kernels, but a Gaussian kernel family is inevitably used in the composition.) The problem with KMS is that it is an NP-Hard problem so there is a gap between what it can detect in theory and what kernel slice can be optimized.
>
> MLW alone is suitable for detecting a single well-localized shift. It is true that it is not competitive for arbitrary shifts, but the multiple landmark version MMLW is competitive (and much faster) than many methods on the Higgs dataset.
>
> We have now run on the CIFAR-10 versus CIFAR-10.1 (v4) used by Liu et al (2020). We ran our methods (MLW, MLW-split, and MMLW with ME power estimate) using the same seeds and running MMD-O again across 10 trials we got a similar power to that reported.
> | ME | SCF | C2ST-S | C2ST-L |  MMD-D | MMD-O| MMD-O (again)|  MLW| MLW-split | MMLW |
> |---|---|---|---|---|---|---|---|---|---|
> | 0.588 | 0.171 | 0.452 | 0.529 |  0.744 | 0.316 | 0.35±0.18|**0.84**±0.15  |0.37±0.33 | 0.50±0.22 |
>
>
> “MLW distance is nothing more than a special case of the KMS distance” while this is correct it does not recognize that computing KMS is not tractable, while MLW and MMLW are tractable. We relate them by an inequality after (3).  Because KMS is hard to compute it is hard to empirically find how much weaker they are. We tried initializing KMS at the optimal landmark slice solution to see if it improved, but this alone did not guarantee that the optimization of the max-min objective was not getting stuck. Nonetheless, this did provide some insight, namely that although KMS is less restricted, the norm constraint means that it is not well motivated to be too diffuse.
>
> In terms of theory, ideally we would show under certain situations where MMLW is more powerful than KMS.  In this case, we must first define the MMLW test statistic as the average of $q$ landmark sliced Wasserstein distance (originally it was the sum). The necessary assumption for proving a MMLW is more powerful is that if each landmark selected by MMLW provides a similar kernel sliced Wasserstein distance that is equal to the KMS, and the threshold based on the quantile under the null hypothesis for MMLW is less than the threshold for KMS. The latter fact follows since the Rademacher complexity is lower. We note that variance of the MMLW test statistic will be lower than KMS as it is the average of $q$. Additionally, the test statistic difference from the threshold will be greater for MMLW since the threshold should necessarily be smaller. Then MMLW will be more powerful than KMS follows from application of Cantelli's inequality.
>
> In terms of interest to the TMLR community, a method that detects localized discrepancies seems very interesting. As noted in our paper, the Liu et al. (2020) ICML paper proposed using ME for interpretation has been cited by 284. Two-sample testing has been applied to detect mode dropping https://arxiv.org/pdf/1806.07755 using a suitable feature space. While we do not show extensive real-world datasets, we highlight key cases were with sufficient sample size you can trust MLW to identify a meaningful discrepancy and MMLW would likewise identify a set of meaningful discrepancies.
>
> **Response to requested changes (to be added to revision):**
> 1. MLW will be weaker than KMS, when the change is not localized to a single landmark. However, if the distribution shift consists of multiple localized discrepancies then MMLW could be stronger.
> 2. See comments above.
> 3. The Gaussian kernel is motivated by its characteristic nature (conditions of Theorem 1) and that it is strictly positive definite and normalized. We test in the appendix the Laplacian kernel too.
> 4. Our sketch of a new theorem shows that the power will be stronger if the change consists of multiple localized discrepancies.
> 5. Will add results for CIFAR-10.1.

---

### Review · Reviewer_7eAj · 2026-03-10

**Summary Of Contributions:**

This paper proposes the Max Landmark Kernel Wasserstein (MLW) distance for detecting distribution differences between two datasets. The key idea is to restrict the slicing functions to kernel embeddings of datapoints (called landmarks in the paper ), enabling a discrete search over candidate landmarks and providing localized interpretation of distribution discrepancies. The authors also propose MMLW, a greedy multi-landmark extension with a diversity regularization term. Experiments on MNIST and CIFAR-10 datasets show the effective of the methods.

**Additional Comments:**

Some discussions of this trade-off with the full-kernel method and its implications would further improve the paper

**Audience:**

Yes

**Audience Explanation:**

Community working in covarite shift, distribution shift, or broadly statistical ML.

**Broader Impact Concerns:**

None noted.

**Claims And Evidence:**

No

**Claims Explanation:**

I think the paper needs to improve in terms of the scope and narrative. Detailed in the requested changes.

**Requested Changes:**

1. The paper frequently frames the method as detecting covariate shift, but the method only tests whether the marginal feature distributions differ, but didn't really worry about whether p(y|x) changes. The terminology should be clarified or softened throughout the paper to avoid overclaiming.

2. The empirical evaluation focuses primarily on class imbalance scenarios, which represent a narrow and structured form of distribution shift. This also posts some overclaiming issues, since the method is only demonstrated on a narrow scope of covariate shift.

3. Evaluating the method under domain adaptation-style covariate shifts (e.g., cross-domain datasets or corruption benchmarks) would better support the claims made in the paper.

---

> ### Author Response · Authors · 2026-03-24
>
> We appreciate the reviewer’s perspectives. We plan to make minor revisions to the paper to clarify the covariate shift setting compared to marginal distributions and plan to have more result; unfortunately, the results are not ready and the response window is adding.
>
> 1. We agree that we simply test whether the marginal features differ, but our analysis is not designed to test whether $p(y|x)$ changes. We note that the test itself doesn’t have access to $y$. However, the synthetically imbalanced datasets are created controlling the distribution over the classes $y$. For example, in the CIFAR-10 versus CIFAR-10.1 introduced by Liu et al. (2020), the authors mention that only checking the marginal distribution is necessary (compared to the joint). One could assume the labeling $p(y|x)$ hasn’t changed. Furthermore, the use of representations from the latent representation of classifier models as in BBSD makes sense for covariate shift detection, because if the relationship between $y$ and $x$ has changed, then looking at the marginal through the lens of latent of a model for $p(y|x)$ does not make sense. Thus, we should clarify that in absence of any suggesting that labeling has changed, a change in marginal distribution of the covariates is covariate shift.
>
> 2. We agree that class imbalance scenarios are a narrow form of distribution shift. But we note that the appendix contains examples of Gaussian distributions, we tested Higgs dataset, and we have now added CIFAR-10 versus CIFAR-10.1. We also have plans to add illustrative examples of MNIST-C (for different corrupted forms); and comparison of test data to synthetic output of generative models.
>
> 3. Cross-domain datasets are an interesting suggestion in addition to the MNIST-C. We plan to look into the Office 31 dataset which has DSLR camera images, web cam images, and images from Amazon.
>
> Regarding the trade-off with the full-kernel method, please see the response to Reviewer EjpB. The other response also has results for CIFAR-10.1.

---

### Review · Reviewer_yGHd · 2026-03-11

**Summary Of Contributions:**

The paper proposes a new method of estimating the difference between two distributions in a localized way, with an application to two sample testing. This idea is based on finding one or more "landmarks" in the data which maximally slice the data in kernel space according to the data points' kernelized projections onto the landmark. They demonstrate that due to the unidimensional nature of this problem, finding the maximizing landmark can be done efficiently. The paper then offers experiments showing the ability of this metric to identify localized covariate shift.

**Audience:**

Yes

**Audience Explanation:**

The idea of a maximal-slice divergence measure that is also provably efficiently calculatable seems a useful tool for future work, and this measure's advantage in complex, high-dimensional data spaces such as those common in deep learning suggest that the method could be useful for covariate shift identification in real-world scenarios.

**Claims And Evidence:**

Yes

**Claims Explanation:**

The metric is well-motivated, clearly defined, and shown to be efficiently calculatable. It is also properly situated within the existing literature. Experiments demonstrate that in many settings this metric outperforms existing ones for two-sample testing.

**Requested Changes:**

In general I think this paper is well-motivated. My only concern is that quite a lot of the earlier discussion introduces much more complex mathematical setup than is actually needed/used in the later method. Several times while reading the paper I was unsure whether it was important for me to take the time to fully internalize the math as it was being presented, or if it was ok for me to just understand the general gist before moving on. This is most prevalent and jarring in Section 2.

I would suggest the paper could be greatly simplified and clarified if the authors shorten the math in Section 2 and refocus it on only the details that will actually be necessary for later sections. The fact that much of the analysis *could* be applied more generally is not necessarily important; if the authors feel it is essential, perhaps they could briefly mention this fact and reference a more detailed discussion in the Appendix. But generally I think the math set up can be greatly simplified and make for a much easier transition into the formal elements of the paper.

---

> ### Author Response · Authors · 2026-03-24
>
> We appreciate the reviewer’s thoughtful evaluation of our paper. We understand the concern that much of Section 2 is not directly necessary for the proposed method; however, Section 2 also provides the context of the most closely related work and baselines. Thus, the section could be more accurately labeled as Preliminaries and Related Work. Nonetheless, in the revision we plan to take steps to shorten and refocus Section 2 as much as possible, keeping only the mathematical setups that will be used in later sections (and baselines), but maintaining verbal descriptions of related work.

---

### Decision · Action_Editor_y6sT · 2026-04-23

**Recommendation:** Reject

**Audience:**

Yes

**Audience Explanation:**

Detecting distribution shifts is a problem that arises in many real-world problems. Therefore, TMLR's audience will have interested in this topic broadly.

**Claims And Evidence:**

No

**Claims Explanation:**

The paper proposes an interpretable and computationally efficient variant of kernel Wasserstein-based two-sample testing via landmark selection. In particular, major contributions of this paper are novel operationalization of kernel slicing via discrete landmarks, having strong interpretability and computational advantages, and some evidence of competitive performance in controlled settings was provided.

While the idea is technically sound and potentially useful, the current submission suffers from overstated claims, insufficient empirical validation, and unclear positioning relative to prior work. In particular the relationship to the kernel max-sliced Wasserstein distance is not adequately clarified and contributions appear incremental without deeper justification. In addition, evaluation is too narrow, does not convincingly support claims, and performance degrades in more realistic scenarios. Furthermore, writing could be improved since the current presentation includes unnecessary theoretical overhead and core contributions are obscured.